# DemoDICE: Offline Imitation Learning with Supplementary Imperfect Demonstrations

**Geon-Hyeong Kim**[1]**, Seokin Seo**[2]**, Jongmin Lee**[1]**, Wonseok Jeon**[†,3,4]**, HyeongJoo Hwang**[2]**,
Hongseok Yang**[1,2,5]**, Kee-Eung Kim**[1,2]

[1]School of Computing, KAIST, Daejeon, Republic of Korea
[2]Kim Jaechul Graduate School of AI, KAIST, Daejeon, Republic of Korea
[3]Mila, Quebec AI Institute
[4]School of Computer Science, McGill University
[5]Discrete Mathematics Group, Institute for Basic Science (IBS), Daejeon, Republic of Korea

## Abstract

We consider offline imitation learning (IL), which aims to mimic the expert's behavior from its demonstration without further interaction with the environment. One of the main challenges in offline IL is to deal with the narrow support of the data distribution exhibited by the expert demonstrations that cover only a small fraction of the state and the action spaces. As a result, offline IL algorithms that rely only on expert demonstrations are very unstable since the situation easily deviates from those in the expert demonstrations. In this paper, we assume additional demonstration data of unknown degrees of optimality, which we call imperfect demonstrations. Compared with the recent IL algorithms that adopt adversarial minimax training objectives, we substantially stabilize overall learning process by reducing minimax optimization to a direct convex optimization in a principled manner. Using extensive tasks, we show that DemoDICE achieves promising results in the offline IL from expert and imperfect demonstrations.

## 1 Introduction

Reinforcement learning (RL) (Sutton et al., 1998) aims to learn a intelligent behavioral strategy based on reward feedback. Although RL has achieved remarkable success in many challenging domains, its practicality and applicability are still limited in two respects. First, we need to specify the reward function, which may be non-trivial to do so in many real-world problems that require complex decision making. Second, the standard RL setting assumes online interaction with the environment during the intermediate stages of learning, which is infeasible for mission-critical tasks.

Imitation learning (IL) (Pomerleau, 1991; Ng & Russell, 2000) addresses the first limitation of RL, where the agent is trained to mimic the expert from demonstration instead of specifying the reward function. It is well known that adopting supervised learning for training the imitating agent, commonly referred to as behavioral cloning (BC), is vulnerable to the distribution drift (Ross et al., 2011). Thus, most of the successful IL algorithms rely on online experiences collected from the environment by executing intermediate policies during training. Recent progress made by adversarial imitation learning (AIL) (Ho & Ermon, 2016; Ke et al., 2019; Kostrikov et al., 2020) achieving state-of-the-art results on challenging imitation tasks still relies on such an online training paradigm.

Unfortunately, in many realistic tasks such as robotic manipulation and autonomous driving, online interactions are either costly or dangerous. Offline RL (Fujimoto et al., 2019; Kumar et al., 2019; 2020; Levine et al., 2020; Wang et al., 2020; Lee et al., 2021; Kostrikov et al., 2021) aims to address these concerns by training the agent from the pre-collected set of experiences without online interactions. To prevent an issue caused by the distributional shift, offline RL algorithms mitigate the phenomenon by constraining the shift or making a conservative evaluation of the policy being learned.

---

[†]The research that is the basis of this paper was done while the author was at Mila/McGill University, but the author is currently employed by Qualcomm Technologies Inc.

In this paper, we are concerned with offline IL problems. Finding an effective algorithm for these problems is tricky. For instance, naively extending the offline RL algorithms (which assume a reward function) to the offline IL setting does not work. In practice, expert demonstrations are scarce due to the high cost of obtaining them. Thus, they typically cover only a small fraction of the state and action spaces, which in turn makes the distribution drift issue even more stand out compared with the standard offline RL setting with a reward function. We mitigate this issue by assuming a large number of supplementary imperfect demonstrations, without requiring any level of optimality for these imperfect demonstrations; they may contain expert or near-expert trajectories (Wu et al., 2019; Wang et al., 2021) as well as non-expert ones all together. This generality covers the situations from real-world applications, but at the same time, it poses a significant challenge for the design of a successful offline IL algorithm.

In this paper, we propose DemoDICE, a novel model-free algorithm for offline IL from expert and imperfect demonstrations. We formulate an offline IL objective which not only mitigates distribution shift from the demonstration-data distribution but also naturally utilizes imperfect demonstrations. Our new formulation allows us to compute a closed form solution, which learns a policy in the space of stationary distributions but suffers from the instability issue in practice. We tackle the issue by proposing an alternative objective, which leads to a stable algorithm in practice while keeping the optimal stationary distribution. Finally, we introduce a method to extract the expert policy from a learned stationary distribution in a simple yet effective way. Our extensive evaluations show that DemoDICE achieves performance competitive to or better than a state-of-the-art off-policy IL algorithm in the offline-IL tasks with expert and imperfect demonstrations. The code to reproduce our results is available at our GitHub repository[1].

## 2 PRELIMINARIES

### 2.1 MARKOV DECISION PROCESS (MDP)

We assume an environment modeled as a Markov Decision Process (MDP), defined by tuple $M = \langle S, A, T, R, p_0, \gamma \rangle$, where $S$ is the set of states, $A$ is the set of actions, $T : S \times A \to \Delta(S)$ is the probability $p(s_{t+1}|s_t, a_t)$ of making transition from state $s_t$ to state $s_{t+1}$ by executing action $a_t$ at timestep $t$, $R : S \times A \to \mathbb{R}$ is the reward function, $p_0 \in \Delta(S)$ is the distribution of the initial state $s_0$, and $\gamma \in [0, 1]$ is the discount factor. A policy $\pi : S \to \Delta(A)$ of MDP $M$ is a mapping from states of $M$ to distributions over actions. For the given policy $\pi$, the stationary distribution $d^\pi$ is defined as follows:

$$d^\pi(s, a) = (1 - \gamma) \sum_{t=0}^{\infty} \gamma^t p\big(s_t = s, a_t = a \big| s_0 \sim p_0(\cdot), s_t \sim T(\cdot|s_{t-1}, a_{t-1}), a_t \sim \pi(\cdot|s_t)\big)$$

We assume a precollected dataset $D^E$ of $(s, a, s')$ tuples generated by the expert, and a precollected imperfect dataset $D^I$ (generated by unknown degrees of optimality). More precisely, for the (underlying) expert policy's stationary distribution $d^E(s, a)$, we assume that $(s, a, s') \in D^E$ is sampled as $(s, a) \sim d^E, s' \sim T(\cdot|s, a)$. We define $D^U = D^E \cup D^I$, the union of two datasets, and denote the corresponding stationary distribution of the dataset $D^U$ as $d^U$. We denote the trajectories generated by expert policy as expert trajectories, and trajectories generated by non-expert policies as non-expert trajectories. The expert demonstrations consist only of expert trajectories and imperfect demonstrations consist of a mixture of expert and non-expert trajectories. In this paper, we assume that the quality of imperfect demonstrations is unknown.

### 2.2 IMITATION LEARNING

Behavior cloning (BC) is a classical IL approach, which attempts to find a function that maps $s$ to $a$ via supervised learning. The standard BC finds a policy $\pi$ by minimizing the negative log-likelihood:

$$\min_\pi J_{\text{BC}}(\pi) := \min_\pi -\frac{1}{|D|} \sum_{(s,a) \in D} \log \pi(a|s). \tag{1}$$

---

However, it is known to be brittle (Ross et al., 2011) when the interaction with the environment deviates from the scarce trajectories in $D^E$. In such cases, BC fails to recover expert policies.

One of the notable approaches for IL is to formulate the problem as distribution matching (Ho & Ermon, 2016; Ke et al., 2019; Kostrikov et al., 2020). When instantiated with the KL divergence widely used in previous IL works (Ke et al., 2019; Kostrikov et al., 2020), the approach amounts to finding a policy $\pi$ by optimizing the following objective:

$$\max_\pi -D_{\mathrm{KL}}(d^\pi \| d^E) = \mathbb{E}_{(s,a) \sim d^\pi} \left[ \log \frac{d^E(s,a)}{d^\pi(s,a)} \right]. \tag{2}$$

Since we cannot directly access the exact value of $d^E(s,a)$ and $d^\pi(s,a)$, we estimate their ratio using samples from $d^E(s,a)$ and $d^\pi(s,a)$, given as follows:

$$\max_{c:S \times A \to (0,1)} \mathbb{E}_{(s,a) \sim d^E}[\log c(s,a)] + \mathbb{E}_{(s,a) \sim d^\pi}[\log(1 - c(s,a))]. \tag{3}$$

Here, the optimal discriminator $c^*$ recovers $\log c^*(s,a) - \log(1 - c^*(s,a)) = \log \frac{d^E(s,a)}{d^\pi(s,a)}$. Based on this connection between generative adversarial networks (GANs) and IL, AIL algorithms focus on recovering the expert policy (Ho & Ermon, 2016; Kostrikov et al., 2019). However, the agent obtain samples from $d^\pi$ through interaction with environment, which is impossible in offline setting. Therefore, to tackle the offline IL problems, we should derive an alternative estimation without on-policy samples.

## 3 DemoDICE

In this section, we present a novel model-free offline IL algorithm named *offline imitation learning using additional imperfect Demonstrations via stationary DIstribution Correction Estimation* (DemoDICE). Starting from a regularized offline IL objective which accords with offline RL algorithms, we present a formulation that does not require on-policy samples. Such formulation allows us to construct a nested maximin optimization for offline IL from expert and imperfect demonstrations (Section 3.1). Then, we derive the closed-form solution to the sub-problem of the aforementioned optimization and obtain a simple convex optimization objective (Section 3.2). Since the objective is unstable in practice, we transform the objective to an alternative yet still convex objective (Section 3.3). Finally, we show how to extract the policy from the learned correction term (Section 3.4).

### 3.1 Transform constrained optimization into nested optimization

In the context of offline RL, most works use expected return maximization with some regularization to overcome the extrapolation error in offline settings (Fujimoto et al., 2019; Kumar et al., 2019; Nachum et al., 2019b; Kumar et al., 2020; Lee et al., 2021). In this work, we use KL divergence minimization between $d^\pi$ and $d^E$ with KL-regularization:

$$\pi^* := \arg\max_\pi -D_{\mathrm{KL}}(d^\pi \| d^E) - \alpha D_{\mathrm{KL}}(d^\pi \| d^U), \tag{4}$$

where $\alpha \geq 0$ is a hyperparameter that controls the balance between minimizing KL divergence with $d^E$ and preventing deviation of $d^\pi$ from $d^U$.

Many online AIL algorithms estimate divergence between expert and current policy using on-policy samples, which is not available in offline scenario. In contrast, to construct a tractable optimization problem in the offline setting, we consider a problem equivalent to Equation 4 in terms of stationary distribution $d$:

$$\max_d \quad -D_{\mathrm{KL}}(d \| d^E) - \alpha D_{\mathrm{KL}}(d \| d^U) \tag{5}$$

$$\text{s.t} \sum_a d(s,a) = (1-\gamma)p_0(s) + \gamma \sum_{\bar{s},\bar{a}} T(s|\bar{s},\bar{a})d(\bar{s},\bar{a}) \ \forall s, \tag{6}$$

$$d(s,a) \geq 0 \ \forall s,a. \tag{7}$$

The constraints (6-7) are called the Bellman flow constraints. The Lagrangian of the above constrained optimization problem is

$$\max_{d \geq 0} \min_{\nu} -D_{\mathrm{KL}}(d\|d^E) - \alpha D_{\mathrm{KL}}(d\|d^U) + \sum_s \nu(s)((1-\gamma)p_0(s) + \gamma(\mathcal{T}_*d)(s) - (\mathcal{B}_*d)(s)), \quad (8)$$

where $\nu(s)$ are the Lagrange multipliers, $(\mathcal{B}_*d)(s) := \sum_a d(s,a)$ is the marginalization operator, and $(\mathcal{T}_*d)(s) := \sum_{\bar{s},\bar{a}} T(s|\bar{s},\bar{a})d(\bar{s},\bar{a})$ is the transposed Bellman operator. We introduce following derivations for the optimization (8) to obtain tractable optimization in the offline setting:

$$- D_{\mathrm{KL}}(d\|d^E) - \alpha D_{\mathrm{KL}}(d\|d^U) + \sum_s \nu(s)((1-\gamma)p_0(s) + \gamma(\mathcal{T}_*d)(s) - (\mathcal{B}_*d)(s))$$

$$= (1-\gamma)\mathbb{E}_{s \sim p_0}[\nu(s)] + \mathbb{E}_{(s,a) \sim d}\left[\gamma(\mathcal{T}\nu)(s,a) - \nu(s) - \log \frac{d(s,a)}{d^E(s,a)} - \alpha \log \frac{d(s,a)}{d^U(s,a)}\right] \quad (9)$$

$$= (1-\gamma)\mathbb{E}_{p_0}[\nu(s)] + \mathbb{E}_d\left[\gamma(\mathcal{T}\nu)(s,a) - \nu(s) + \underbrace{\log \frac{d^E(s,a)}{d^U(s,a)}}_{:=r(s,a)} - (1+\alpha)\underbrace{\log \frac{d(s,a)}{d^U(s,a)}}_{:=w(s,a)}\right], \quad (10)$$

the equality in Equation 9 holds from the following properties of transpose operators:

$$\sum_s \nu(s)(\mathcal{B}_*d)(s) = \sum_{s,a} d(s,a)(\mathcal{B}\nu)(s,a) \quad \text{and} \quad \sum_s \nu(s)(\mathcal{T}_*d)(s) = \sum_{s,a} d(s,a)(\mathcal{T}\nu)(s,a),$$

where $(\mathcal{B}\nu)(s,a) = \nu(s)$, $(\mathcal{T}\nu)(s,a) = \sum_{s'} T(s'|s,a)\nu(s')$, with assumption $d^E(s,a) > 0$ when $d(s,a) > 0$ (Nachum et al., 2019a). We introduce another log ratio, denoted by $r(s,a)$ in Equation 10 to avoid using $\log \frac{d^E(s,a)}{d(s,a)}$, which requires on-policy samples to estimate. Unlike $\log \frac{d^E(s,a)}{d(s,a)}$, we can estimate $r(s,a)$ in the offline setting using $d^E$ and $d^U$, as we will discuss in the next section in detail.

We change the distribution used in the expectation of Equation 10 from $d$ to $d^U$ by following the standard trick of importance sampling as follows:

$$(1-\gamma)\mathbb{E}_{s \sim p_0}[\nu(s)] + \mathbb{E}_{(s,a) \sim d}[\underbrace{r(s,a) + \gamma(\mathcal{T}\nu)(s,a) - (\mathcal{B}\nu)(s,a)}_{:=A_\nu(s,a)(\text{advantage using } \nu)} - (1+\alpha)\log w(s,a)]$$

$$= (1-\gamma)\mathbb{E}_{s \sim p_0}[\nu(s)] + \mathbb{E}_{(s,a) \sim d^U}\left[w(s,a)\big(A_\nu(s,a) - (1+\alpha)\log w(s,a)\big)\right]$$

$$=: L(w,\nu;r). \quad (11)$$

As an alternative, one can convert expectation of $d$ to $d^E$ instead of $d^U$ in Equation 10 by the similar application of the trick of importance sampling.

$$(1-\gamma)\mathbb{E}_{s \sim p_0}[\nu(s)] + \mathbb{E}_{(s,a) \sim d^E}\left[\exp(-r(s,a))w(s,a)\big(A_\nu(s,a) - (1+\alpha)\log w(s,a)\big)\right].$$

In practice, due to the small number of demonstrations in $d^E$, there may be a lack of diversity. To release this practical issue, we prefer to use $d^U$ instead of $d^E$.

In summary, DemoDICE solves the following maximin optimization:

$$\max_{w \geq 0} \min_{\nu} L(w,\nu;r), \quad (12)$$

where $r$ is trained by using precollected datasets. Note that the solution $w^*$ of Equation 12 with the ground-truth ratio $r$ is the ratio of two distributions, the stationary distribution $d^{\pi^*}$ of the expert policy $\pi^*$ and the stationary distribution $d^U$ of union of expert and imperfect demonstrations.

## 3.2 PRETRAINED STATIONARY DISTRIBUTION RATIO AND A CLOSED-FORM SOLUTION

To solve the problem (12), we can pretrain $r$, the log-ratio of $d^E(s,a)$ and $d^U(s,a)$. To this end, we train a discriminator $c : S \times A \to [0,1]$ using the following maximization objective:

$$\max_{c:S \times A \to [0,1]} J_c(d^E, d^U) := \mathbb{E}_{d^E}[\log c(s,a)] + \mathbb{E}_{d^U}[\log(1 - c(s,a))], \quad (13)$$

whose maximizer is $c^*(s,a) = \frac{d^E(s,a)}{d^U(s,a)+d^E(s,a)}$. By using $c^*$, $r$ can be also obtained as

$$r(s,a) = -\log\left(\frac{1}{c^*(s,a)} - 1\right). \tag{14}$$

Since the optimization (8) is a convex optimization and the strong duality holds, the optimization (8) is equal to

$$\min_\nu \max_{w\geq 0} L(w,\nu;r). \tag{15}$$

The closed-form solution of inner max optimization of (15) turns out to be $w_\nu^*(s,a) = \exp\left(\frac{A_\nu(s,a)}{1+\alpha} - 1\right)$ for all $(s,a)$, where $A_\nu(s,a) := r(s,a) + \gamma(\mathcal{T}\nu)(s,a) - \nu(s)$. By using $w_\nu^*(s,a)$, optimization (15) can be reduced to a simple convex minimization problem as follows:

$$\min_\nu L(w_\nu^*,\nu;r) = (1-\gamma)\mathbb{E}_{s\sim p_0}[\nu(s)] + (1+\alpha)\mathbb{E}_{(s,a)\sim d^U}\left[\exp\left(\frac{A_\nu(s,a)}{1+\alpha} - 1\right)\right] \tag{16}$$

Although $L(w_\nu^*,\nu;r)$ in Equation 16 is convex on $\nu$, we observe that $L(w_\nu^*,\nu;r)$ in Equation 16 is numerically unstable in practice. Especially, the exponential term in $L(w_\nu^*,\nu;r)$ is prone to explosion and so is the gradient $\nabla_\nu L(w_\nu^*,\nu;r)$.

### 3.3 Alternative convex optimization

In order to derive a numerically-stable alternative objective, we describe the following theoretical result:

**Proposition 1.** *Define the objective $\widetilde{L}(\nu;r)$ as*

$$\widetilde{L}(\nu;r) := (1-\gamma)\mathbb{E}_{s\sim p_0}[\nu(s)] + (1+\alpha)\log\mathbb{E}_{(s,a)\sim d^U}\left[\exp\left(\frac{A_\nu(s,a)}{1+\alpha}\right)\right]. \tag{17}$$

*Then, the following equality holds:*

$$\min_\nu L(w_\nu^*,\nu;r) = \min_\nu \widetilde{L}(\nu;r).$$

*(The proof can be found in Appendix A.)*

The objective $L(w_\nu^*,\nu;r)$ in Equation 16 has the instability issue since the gradient with respect to $\nu$ involves an unbounded easily-exploding function $\exp(\cdot)$. In contrast, the alternative objective $\widetilde{L}(\nu;r)$ in Equation 17 does not suffer from the same stability issue because in this case, the gradient forms a soft-max and is bounded by 1. Furthermore, we observe that the surrogate objective is still convex:

**Proposition 2.** *The objective $\widetilde{L}(\nu;r)$ is convex with respect to $\nu$. (The proof is in Appendix C.)*

To summarize, we have presented a training objective $\widetilde{L}(\nu;r)$ which significantly stabilizes the objective $L(w_\nu^*,\nu;r)$ while having the same optimal value. We converted the minimax optimization (15) to a single minimization, which is much more stable. Proposition 1 is, however, limited in that it is not explain how to sample actions from learned $\tilde{\nu}$. Thus, we need a method to sample action from learned $\tilde{\nu}$ and the next section presents such a method.

### 3.4 Policy extraction

In this section, we present a method for extracting a policy from the optimal $\tilde{\nu}^* = \arg\min_\nu \widetilde{L}(\nu;r)$. Our method is based on the following optimization, which amounts to the variant of weighted BC:

$$\min_\pi -\mathbb{E}_{(s,a)\sim d^{\pi^*}}[\log\pi(a|s)] = -\mathbb{E}_{(s,a)\sim d^U}[w^*(s,a)\log\pi(a|s)], \tag{18}$$

where $w^*(s,a) = \frac{d^{\pi^*}(s,a)}{d^U(s,a)}$. It can be estimated by $\nu^* \in \arg\min_\nu L(w_\nu^*,\nu;r)$, but we use $\widetilde{L}(\nu;r)$ in Equation 17 to train $\nu$. Here, from the Proposition 1, we introduce theoretical results to explain the relation between $\nu^* \in \arg\min_\nu L(w_\nu^*,\nu;r)$ and $\tilde{\nu}^* \in \arg\min_{\tilde{\nu}} L(\tilde{\nu};r)$:

**Corollary 1.** *Let* $\widetilde{V} = \arg\min_\nu \widetilde{L}(\nu; r)$ *and* $V = \arg\min_\nu L(w_\nu^*, \nu; r)$ *be the sets of* $\nu$. *Then the following equation holds: (We provide the proof in Appendix B.)*

$$\widetilde{V} = \{\nu + c | \nu \in V, c \in \mathbb{R}\} \tag{19}$$

We observe that for any $\tilde{\nu}^* \in \widetilde{V}$, there are $\nu^* \in V$ and $c \in \mathbb{R}$ such that $\tilde{\nu}^* = \nu^* + c$. Based on this relationship, we can easily derive the following proportion: (see Appendix D for more detailed derivation)

$$\widetilde{w}_{\tilde{\nu}^*}(s, a) := \exp\left(\frac{A_{\tilde{\nu}^*}(s, a)}{1 + \alpha}\right) \propto \frac{d^*(s, a)}{d^U(s, a)}.$$

In other words, using $\tilde{\nu}^* \in \widetilde{V}$, we can only obtain an unnormalized stationary distribution ratio instead of exact ratio. Therefore, we estimate Equation 18 using self-normalized importance sampling (Owen, 2013) as shown below:

$$\min_\pi J_\pi(\tilde{\nu}^*) = -\frac{\mathbb{E}_{(s,a)\sim d^U}\left[\widetilde{w}_{\tilde{\nu}^*}(s, a) \log \pi(a|s)\right]}{\mathbb{E}_{(s,a)\sim d^U}\left[\widetilde{w}_{\tilde{\nu}^*}(s, a)\right]}. \tag{20}$$

Based on the weighted BC, we simply extract the policy from $\tilde{\nu}^* \in \widetilde{V}$. Our policy extraction is highly related to variants of weighted BC (Wang et al., 2018; 2020; Siegel et al., 2020) in offline RL. The common idea of these methods is (1) training an action-value function $Q(s, a)$ and (2) estimating the corresponding advantage $A(s, a)$. Using an increasing, non-negative function $f(\cdot)$ (*e.g.,* $\exp(\cdot)$), they perform weighted BC on precollected dataset with defining weight $f(A(s, a), s, a)$ as:

$$\arg\min_\pi \mathbb{E}_{(s,a)\sim d^U}\left[-f(A(s, a), s, a) \log \pi(a|s)\right].$$

Finally, if we regard the advantage of optimal $\nu^*$ in Equation 16 as an advantage function, our policy extraction matches the offline RL based on weighted BC.

## 4 RELATED WORKS

**Learning from imperfect demonstrations**  Among several recent works (Wu et al., 2019; Brown et al., 2019; 2020; Brantley et al., 2020; Tangkaratt et al., 2020; Sasaki & Yamashina, 2021; Wang et al., 2021) that attempted to leverage imperfect demonstrations in IL, two-step importance weighting imitation learning (2IWIL), generative adversarial imitation learning with imperfect demonstration and confidence (IC-GAIL) (Wu et al., 2019), weighted GAIL (WGAIL) (Wang et al., 2021), and Behavioral Cloning from Noisy Demonstrations (BCND) (Sasaki & Yamashina, 2021) are closely related to our work.

2IWIL, IC-GAIL and WGAIL suppose imperfect demonstrations which are composed of expert and non-expert trajectories and this definition is the same as that of DemoDICE. Assuming that some of trajectories in the imperfect demonstrations are provided with labels of confidence of optimalities, 2IWIL trains a semi-supervised classifier to predict confidence scores for unlabeled imperfect demonstrations and train the policy using the score. Instead of such two-step learning, IC-GAIL trains the policy in an end-to-end fashion.

Inspired from Wu et al. (2019), Wang et al. (2021) suggest a general weighted GAIL objective function. Unlike previous work, WGAIL predicts the weight for imperfect demonstrations without information about their quality. However, 2IWIL, IC-GAIL, and WGAIL are online on-policy imitation learning method, whereas our algorithm, DemoDICE, is an offline imitation learning method.

BCND assumes that the expert demonstrations are sampled from a noisy expert policy. BCND learns ensemble policies with a weighted BC objective, where the weight is policy learned by weighted BC in the previous internal iteration. As the learning progresses, the mode of the learned policy moves towards that of the noisy expert policy. Thus, the learned policy tends to converge to a non-expert policy when the non-expert trajectories comprise the majority of demonstrations.

**Stationary distribution corrections** In RL and IL, some prior works have used distribution corrections. In AlgaeDICE (Nachum et al., 2019b), regularization in the space of stationary distribution is augmented to a policy optimization objective to solve off-policy RL problems. In addition, by using dual formulation of f-divergence and change of variables (Nachum et al., 2019a), an off-policy learning objective is derived in AlgaeDICE. ValueDICE (Kostrikov et al., 2020) minimizes the KL divergence between the agent and the expert stationary distributions to solve IL problems.

Similar to the AlgaeDICE, ValueDICE obtains an off-policy learning objective for distribution matching by using the dual formulation of KL divergence and change of variables. Both AlgaeDICE and ValueDICE objectives are optimized by nested optimization, which may suffer from numerical instability.

In contrast, OptiDICE (Lee et al., 2021) reduces the same optimization used in AlgaeDICE to unconstrained convex optimization. Although it shows promising performance in offline RL tasks, directly applying it to offline IL from expert and imperfect demonstrations is not trivial as we discussed in Section 3.

Off policy learning from observations (OPOLO) (Zhu et al., 2020) addresses off-policy IL using expert demonstrations composed only of states. It has similarities to DemoDICE in two aspects. Firstly, OPOLO adopts a discriminator to estimate a state-transition distribution ratio. Furthermore, DICE-like technique is applied to the derivation of the objective of OPOLO, which is similar to ours. Thus, based on the relation between DemoDICE and OPOLO, offline IL using expert demonstrations containing only states can be considered as a promising direction for future work.

## 5 EXPERIMENTS

In this section, we present the empirical performance of DemoDICE and baseline methods on MuJoCo continuous control environments (Todorov et al., 2012) using the OpenAI Gym (Brockman et al., 2016) framework. We provide experimental results for 4 MuJoCo environments: Hopper, Walker2d, HalfCheetah, and Ant. We utilize D4RL datasets (Fu et al., 2020) to construct expert and imperfect demonstrations for our experiments. We construct datasets, baselines, and evaluation protocol according to the following procedures:

**Datasets** For each of MuJoCo environments, we utilize three types of D4RL datasets (Fu et al., 2020), whose name end with "-expert-v2", "-full_replay-v2", or "-random-v2". In the remainder of this section, we refer them by using corresponding suffixes. We regard the trajectories in `expert-v2` as expert trajectories. The set of expert demonstrations $D^E$ consists of the first trajectory in `expert-v2`.

**Baselines** We compare our method with three strong baseline methods, BC, BCND (Sasaki & Yamashina, 2021), and ValueDICE (Kostrikov et al., 2020). To consider the potential benefit of utilizing $D^I$, we carefully tuned BC with 5 different values of $\beta \in \{1, 0.75, 0.5, 0.25, 0\}$, which controls the balance between minimizing negative log-likelihood of $D^E$ and minimizing that of $D^U = D^E \cup D^I$ as follows:

$$\min_{\pi} J_{BC(\beta)}(\pi) := -\beta \cdot \frac{1}{|D^E|} \sum_{(s,a) \in D^E} \log \pi(a|s) - (1-\beta) \cdot \frac{1}{|D^U|} \sum_{(s,a) \in D^U} \log \pi(a|s).$$

We denote those 5 different settings by BC($\beta$=1), BC($\beta$=0.75), BC($\beta$=0.5), BC($\beta$=0.25), BC($\beta$=0). For BCND, we use all demonstrations as noisy expert demonstrations and report statistics of the best performance among 5 internal iterations over 5 seeds. We provide the details for BCND in Appenedix E.1. Lastly, we made ValueDICE to utilize imperfect datasets by plugging in all the demonstrations to replay buffer.

**Evaluation metric** The normalized scores for each environment are measured by `normalized score = 100 × `$\frac{\text{score}-\text{random score}}{\text{expert score}-\text{random score}}$, where expert and random scores are average returns of trajectories in `expert-v2` and `random-v2`, respectively. We compute the average normalized score and the standard error over five random seeds. In the following subsections, we provide experimental results on two types of imperfect datasets.

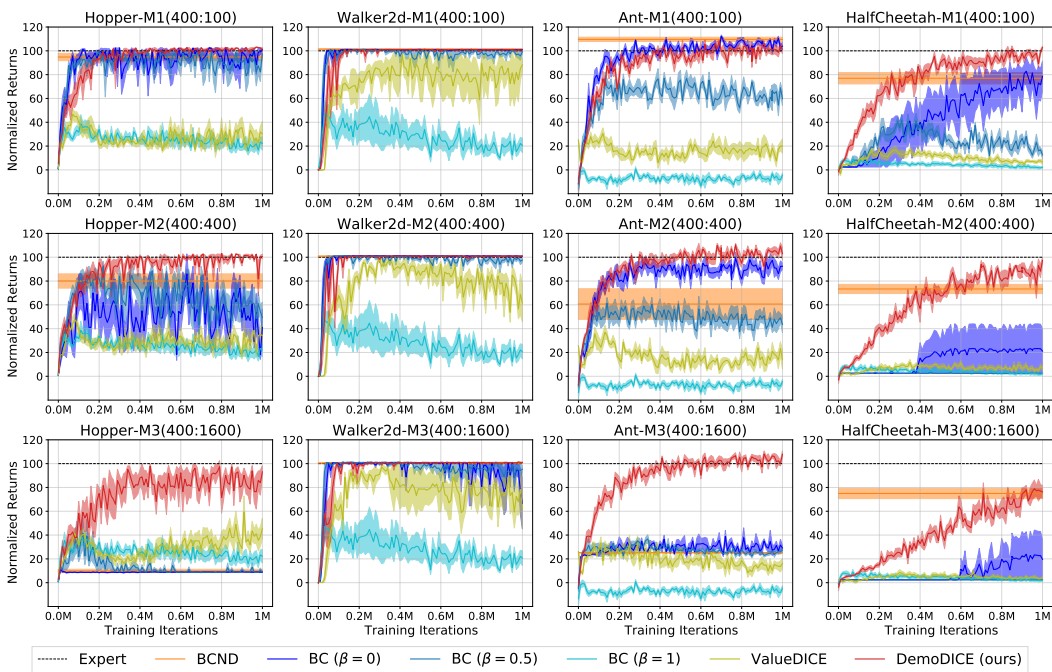

Figure 1: Performance of DemoDICE and baseline algorithms on mixed-dataset tasks M1, M2, and M3. Especially, in the M3 task, which contains a large number of bad trajectories, DemoDICE maintains performance while ValueDICE, BCND, and BC fail to achieve competitive performance. We plot the mean and the standard errors (shaded area) of the normalized scores over 5 random seeds.

## 5.1 MIXED DATASET

DemoDICE mainly aims to overcome distributional drift caused by the lack of expert demonstrations. When DemoDICE successfully learns the optimal unnormalized stationary distribution ratio, we can expect that DemoDICE distinguishes expert trajectories from non-expert ones to update its own policy. Based on this intuition, we hypothesize that if the same sufficiently-many expert trajectories are included in imperfect demonstrations, DemoDICE achieves the optimal performance invariant to the number of sub-expert trajectories in imperfect demonstrations. To see if it is the case, we use mixed datasets which have the same expert trajectories but have different numbers of non-expert trajectories in imperfect demonstrations.

**Experimental setup** Across all environments, we consider 3 tasks, each of which is called one of M1, M2, or M3 and is provided with expert and imperfect demonstrations. While sharing the expert demonstrations composed of the same single expert trajectory, imperfect demonstrations in each task are composed of expert and random trajectories with different ratios such as 1:0.25 (M1), 1:1 (M2), and 1:4 (M3). Specifically, imperfect demonstrations in M1, M2, and M3 are composed of 400 expert trajectories sampled from expert-v2 and 100, 400, and 1600 random trajectories sampled from random-v2, respectively.

**Result** We report the result of DemoDICE and comparing methods in Figure 1. For simplicity, $BC(\beta = 0.25)$ and $BC(\beta = 0.75)$ are in Appendix F. Note that imperfect demonstrations in the tasks M1, M2, and M3 share the same expert trajectories, while having the different number of random trajectories. Ideally, if an algorithm uses only expert trajectories from imperfect demonstrations, the algorithm should have the same performance in the tasks M1, M2, and M3, as discussed at the beginning of this subsection, except convergence speed. While ValueDICE, BCND, and all the variants of BC catastrophically fail to recover expert policies in task M3 on all the environments except Walker2d, DemoDICE reaches the expert performance regardless of environments and tasks.

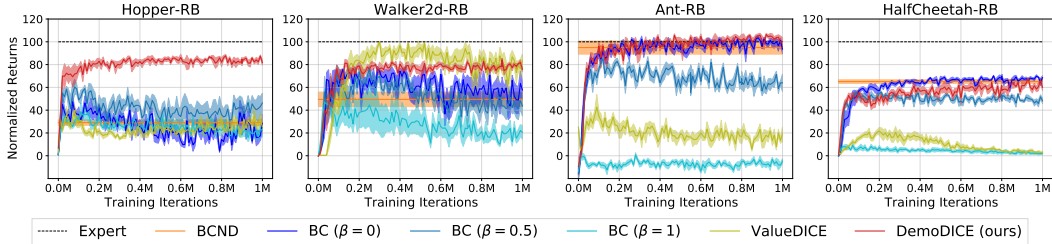

Figure 2: Performance of DemoDICE and baseline algorithms for replay buffer task RB. In RB tasks, DemoDICE achieves better or competitive performances compared to the other baselines. We plot the mean and the standard error (shaded area) of the normalized scores over 5 random seeds.

The result strongly implies that DemoDICE succeeds to learn appropriate stationary distribution correction, which is possible only when the algorithm effectively leverages expert trajectories in imperfect demonstrations ignoring the rest. Furthermore, it is remarkable that DemoDICE reaches the expert performance in HalfCheetah in the end, which can be found in Appendix F.1. To summarize, we have empirically shown that DemoDICE is robust to any choices of the environments and tasks and significantly outperforms existing methods on offline IL.

## 5.2 REPLAY BUFFER DATASET

When we use mixed datasets as imperfect demonstration sets, ValueDICE poorly performs in all environments except Walker2d as we observed in Section 5.1. Since `full_replay-v2` consists of the replay buffer's data during off-policy training, we believe it is one of the most practically-relevant datasets to fill ValueDICE's replay buffer. Therefore, we conduct additional experiments using this dataset as the imperfect demonstrations.

**Experimental setup** Applying the same expert demonstrations used in Section 5.1, we employ another task named RB, which uses `full_replay-v2` directly as its imperfect dataset.

**Result** Figure 2 summarizes the result of DemoDICE and baseline methods. We observe that DemoDICE performs on par with ValueDICE in Walker2d. However, on the other environments, DemoDICE strictly outperforms ValueDICE. We also observe that DemoDICE performs on par with BCND in Ant, HalfCheetah environments, and significantly outperforms in Hopper and Walker2d environments. While showing the performance competitive to the best performing baseline methods in Walker2d, Ant, and HalfCheetah, DemoDICE outperforms all the methods in Hopper by a significant margin.

Since the replay buffer dataset is collected by the policy which evolves (in policy training phase), it can be regarded as a set of imperfect demonstration from a wide range of sources. We emphasize that DemoDICE performs well not only when bad trajectories make up majority of the given imperfect demonstrations but also when they are generated from multiple sources.

## 6 CONCLUSION

We have presented DemoDICE, an algorithm for offline IL from expert and imperfect demonstrations that achieves state-of-the-art performance on various offline IL tasks. We first introduced a regularized offline IL objective and reformulated the objective so as to make it natural to compute closed-form solution. We then tackled the instability coming from the naive application of closed-form solution by the alternative objective which yields not only the same optimal value but also a stable convex optimization. Furthermore, we presented the method to extract an optimal policy with simple weighted BC. Lastly, our extensive empirical evaluations showed that DemoDICE achieves remarkable performance close to the optimal by exploiting imperfect demonstrations effectively.

ACKNOWLEDGMENTS

This work was supported by the National Research Foundation (NRF) of Korea (NRF-2019R1A2C1087634, NRF-2021M3I1A1097938) and Institute of Information & communications Technology Planning & Evaluation (IITP) grant funded by the Korea government(MSIT) (No.2019-0-00075, No.2020-0-00940, No.2021-0-02068) HY was supported by the Engineering Research Center Program through the National Research Foundation of Korea (NRF) funded by the Korean Government MSIT (NRF-2018R1A5A1059921) and also by the Institute for Basic Science (IBS-R029-C1).

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

# A    PROOF OF PROPOSITION 1

We provide the detailed proofs based on the previous works (Nachum & Dai, 2020; Lee et al., 2021). Therefore, we highly recommend to read the works for a better understanding. We first provide a lemma to prove the proposition:

**Lemma 1** (Fenchel duality). *Consider a primal problem given by*

$$\min_{x\in\Omega} J_P(x) := g(x) + h(Ax),$$

*where $g, h : \Omega \to \mathbb{R}$ are convex, lower-semi-continuous and $A$ is a linear operator. Then, the dual of this problem is given by*

$$\max_{y\in\Omega^*} J_D(y) := -g_*(-A_*y) - h_*(y),$$

*where $A_*$ is a (Hermitian) adjoint operator of $A$, i.e., $\langle y, Ax \rangle = \langle A_*y, x \rangle$ for all $x, y$.*

*Proof.* We borrow the proof steps from Nachum & Dai (2020). Remark that the definition of conjugate function of a function $f$:

$$f_*(y) := \max_x \langle x, y \rangle - f(x),$$

where $\langle \cdot, \cdot \rangle$ is inner product. Based on the definition, we can easily derive the following equations:

$$\begin{aligned}
\min_x g(x) + h(Ax) &= \min_x \max_y g(x) + \langle y, Ax \rangle - h_*(y) \\
&= \max_y \left\{ \min_x g(x) + \langle y, Ax \rangle \right\} - h_*(y) \\
&= \max_y \left\{ -\max_x \langle -A_*y, x \rangle - g(x) \right\} - h_*(y) \\
&= \max_y -g_*(-A_*y) - h_*(y).
\end{aligned}$$

$\square$

Now, we prove the proposition:

**Proposition 1.** *Define the objective $\widetilde{L}(\nu; r)$ as*

$$\widetilde{L}(\nu; r) := (1-\gamma)\mathbb{E}_{s\sim p_0}[\nu(s)] + (1+\alpha)\log\mathbb{E}_{(s,a)\sim d^U}\left[\exp\left(\frac{A_\nu(s,a)}{1+\alpha}\right)\right]. \tag{17}$$

*Then, the following equality holds:*

$$\min_\nu L(w_\nu^*, \nu; r) = \min_\nu \widetilde{L}(\nu; r).$$

*(The proof can be found in Appendix A.)*

*Proof.* We first define two functions:

$$\begin{aligned}
g(\cdot) &:= \delta_{\{(1-\gamma)p_0\}}(\cdot), \\
h(\cdot) &:= \langle -r, \cdot \rangle + (1+\alpha)D_{\mathrm{KL}}(\cdot \| d^U),
\end{aligned}$$

where

$$r(s,a) := \log\frac{d^E(s,a)}{d^U(s,a)},$$

$$\delta_C(x) := \begin{cases} 0 & x \in C, \\ \infty & \text{otherwise.} \end{cases}$$

Then, the corresponding conjugate functions are given by

$$g_*(\cdot) := (1-\gamma)\mathbb{E}_{p_0}[\cdot],$$

$$h_*(\cdot) := (1+\alpha)\log\mathbb{E}_{d^U}\left[\exp\left(\frac{\cdot + r}{1+\alpha}\right)\right].$$

Now, we rewrite the optimization problem (5-7) as follows:

$$\max_d -\delta_{\{(1-\gamma)p_0\}}\big(-(\gamma\mathcal{T}-\mathcal{B})_*d\big) - D_{\mathrm{KL}}(d\|d^E) - \alpha D_{\mathrm{KL}}(d\|d^U)$$

$$= -\delta_{\{(1-\gamma)p_0\}}\big(-(\gamma\mathcal{T}-\mathcal{B})_*d\big) - \mathbb{E}_{(s,a)\sim d}[-r(s,a)] - (1+\alpha)D_{\mathrm{KL}}(d\|d^U)$$

$$= -g\big(-(\gamma\mathcal{T}-\mathcal{B})_*d\big) - h(d).$$

By Lemma 1, we reformulate the optimization problem as:

$$\max_d -g\big(-(\gamma\mathcal{T}-\mathcal{B})_*d\big) - h(d)$$

$$\equiv \min_\nu g_*(\nu) + h_*((\gamma\mathcal{T}-\mathcal{B})\nu)$$

$$= (1-\gamma)\mathbb{E}_{p_0}[\nu] + (1+\alpha)\log\mathbb{E}_{d^U}\left[\exp\left(\frac{(\gamma\mathcal{T}-\mathcal{B})\nu+r}{1+\alpha}\right)\right]$$

$$= (1-\gamma)\mathbb{E}_{s\sim p_0}[\nu(s)] + (1+\alpha)\log\mathbb{E}_{(s,a)\sim d^U}\left[\exp\left(\frac{\gamma(\mathcal{T}\nu)(s,a)-(\mathcal{B}\nu)(s,a)+r(s,a)}{1+\alpha}\right)\right]$$

$$= (1-\gamma)\mathbb{E}_{s\sim p_0}[\nu(s)] + (1+\alpha)\log\mathbb{E}_{(s,a)\sim d^U}\left[\exp\left(\frac{r(s,a)+\gamma(\mathcal{T}\nu)(s,a)-\nu(s)}{1+\alpha}\right)\right]$$

$$= (1-\gamma)\mathbb{E}_{s\sim p_0}[\nu(s)] + (1+\alpha)\log\mathbb{E}_{(s,a)\sim d^U}\left[\exp\left(\frac{A_\nu(s,a)}{1+\alpha}\right)\right]$$

It means that $\min_\nu L(w_\nu^*,\nu;r)$ and $\min_\nu \widetilde{L}(\nu;r)$ are the same optimization problems. $\qquad\square$

We obtain the conjugate function $g_*$ using the fact that $\langle a,y\rangle$ is the conjugate of $\delta_a(x)$. We also compute $h_*$ using the two facts: $b\cdot f_*(\frac{y-a}{b})$ is the conjugate of $\langle a,x\rangle + b\cdot f(x)$ and $\log\mathbb{E}_p[\exp(y)]$ is the conjugate of $D_{\mathrm{KL}}(x\|p)$.

## B PROOF OF COROLLARY 1

We remind the objectives in Equation 16 and Equation 17:

$$L(w_\nu^*, \nu; r) := (1 - \gamma)\mathbb{E}_{s \sim p_0}[\nu(s)] + (1 + \alpha)\mathbb{E}_{(s,a) \sim d^U}\left[ \exp\left( \frac{A_\nu(s,a)}{1 + \alpha} - 1 \right) \right],$$

$$\widetilde{L}(\nu; r) := (1 - \gamma)\mathbb{E}_{s \sim p_0}[\nu(s)] + (1 + \alpha)\log \mathbb{E}_{(s,a) \sim d^U}\left[ \exp\left( \frac{A_\nu(s,a)}{1 + \alpha} \right) \right],$$

where

$$A_\nu(s,a) := r(s,a) + \gamma(\mathcal{T}\nu)(s,a) - \nu(s),$$

$$w_\nu^*(s,a) := \exp\left( \frac{A_\nu(s,a)}{1 + \alpha} - 1 \right).$$

Note that for any constant $C$, the advantage for $\nu + C$ is:

$$\begin{aligned}
A_{(\nu+C)}(s,a) &= r(s,a) + \gamma(\mathcal{T}(\nu + C))(s,a) - (\nu(s) + C) \\
&= r(s,a) + \gamma((\mathcal{T}\nu)(s,a) + C) - (\nu(s) + C) \\
&= r(s,a) + \gamma(\mathcal{T}\nu)(s,a) - \nu(s) - (1 - \gamma)C \\
&= A_\nu(s,a) - (1 - \gamma)C
\end{aligned}$$

**Lemma 2.** *For arbitrary function $\nu$ and constant $C$, the equality*

$$\widetilde{L}(\nu; r) = \widetilde{L}(\nu + C; r)$$

*holds.*

*Proof.* From the definition of $\widetilde{L}(\nu; r)$, we derive the following equation:

$$\begin{aligned}
&\widetilde{L}(\nu + C; r) \\
&= (1 - \gamma)\mathbb{E}_{s \sim p_0}[\nu(s) + C] + (1 + \alpha)\log \mathbb{E}_{(s,a) \sim d^U}\left[ \exp\left( \frac{A_{(\nu+C)}(s,a)}{1 + \alpha} \right) \right] \\
&= (1 - \gamma)\mathbb{E}_{s \sim p_0}[\nu(s) + C] + (1 + \alpha)\log \mathbb{E}_{(s,a) \sim d^U}\left[ \exp\left( \frac{A_\nu(s,a) - (1 - \gamma)C}{1 + \alpha} \right) \right] \\
&= (1 - \gamma)\mathbb{E}_{s \sim p_0}[\nu(s) + C] + (1 + \alpha)\log \left[ \exp\left( -\frac{(1 - \gamma)C}{1 + \alpha} \right) \mathbb{E}_{(s,a) \sim d^U}\left[ \exp\left( \frac{A_\nu(s,a)}{1 + \alpha} \right) \right] \right] \\
&= (1 - \gamma)\mathbb{E}_{s \sim p_0}[\nu(s) + C] + (1 + \alpha)\left[ \log \mathbb{E}_{(s,a) \sim d^U}\left[ \exp\left( \frac{A_\nu(s,a)}{1 + \alpha} \right) \right] - \frac{(1 - \gamma)C}{1 + \alpha} \right] \\
&= (1 - \gamma)\mathbb{E}_{s \sim p_0}[\nu(s)] + (1 + \alpha)\log \mathbb{E}_{(s,a) \sim d^U}\left[ \exp\left( \frac{A_\nu(s,a)}{1 + \alpha} \right) \right] \\
&= \widetilde{L}(\nu; r)
\end{aligned}$$

$\square$

**Lemma 3.** *For any function $\nu$, the following inequality always holds:*

$$L(w_\nu^*, \nu; r) \geq \widetilde{L}(\nu; r).$$

*Equality holds if and only if*

$$\mathbb{E}_{(s,a) \sim d^U}\left[ \exp\left( \frac{A_\nu(s,a)}{1 + \alpha} - 1 \right) \right] = 1.$$

*Proof.* For any $y \geq 0$,

$$y - 1 \geq \log y,$$

and equality holds if and only if $y = 1$. From this inequality, we can derive the inequality directly:

$$\mathbb{E}_{(s,a)\sim d^U}\left[\exp\left(\frac{A_\nu(s,a)}{1+\alpha} - 1\right)\right] - 1 \geq \log \mathbb{E}_{(s,a)\sim d^U}\left[\exp\left(\frac{A_\nu(s,a)}{1+\alpha} - 1\right)\right],$$

and equality holds if and only if

$$\mathbb{E}_{(s,a)\sim d^U}\left[\exp\left(\frac{A_\nu(s,a)}{1+\alpha} - 1\right)\right] = 1.$$

Finally, we obtain the following results:

$$L(w_\nu^*, \nu; r) = (1-\gamma)\mathbb{E}_{s\sim p_0}[\nu(s)] + (1+\alpha)\mathbb{E}_{(s,a)\sim d^U}\left[\exp\left(\frac{A_\nu(s,a)}{1+\alpha} - 1\right)\right]$$

$$\geq (1-\gamma)\mathbb{E}_{s\sim p_0}[\nu(s)] + (1+\alpha)\left\{\log\mathbb{E}_{(s,a)\sim d^U}\left[\exp\left(\frac{A_\nu(s,a)}{1+\alpha} - 1\right)\right] + 1\right\}$$

$$\geq (1-\gamma)\mathbb{E}_{s\sim p_0}[\nu(s)] + (1+\alpha)\log\left\{\mathbb{E}_{(s,a)\sim d^U}\left[\exp\left(\frac{A_\nu(s,a)}{1+\alpha} - 1\right)\right]\exp(1)\right\}$$

$$= (1-\gamma)\mathbb{E}_{s\sim p_0}[\nu(s)] + (1+\alpha)\log\mathbb{E}_{(s,a)\sim d^U}\left[\exp\left(\frac{A_\nu(s,a)}{1+\alpha}\right)\right]$$

$$= \widetilde{L}(\nu; r).$$

The equality holds if and only if

$$\mathbb{E}_{(s,a)\sim d^U}\left[\exp\left(\frac{A_\nu(s,a)}{1+\alpha} - 1\right)\right] = 1$$

□

**Lemma 4.** *For any optimal solution $\tilde{\nu}^* = \arg\min_\nu \widetilde{L}(\nu; r)$, there is a constant $C$ such that $\tilde{\nu}^* + C$ is an optimal solution of $\min_\nu L(w_\nu^*, \nu; r)$.*

*Proof.* Let $\tilde{\nu}^*$ be an optimal solution of $\arg\min_\nu \widetilde{L}(\nu; r)$ and

$$C^* := \frac{1+\alpha}{1-\gamma}\log\mathbb{E}_{(s,a)\sim d^U}\left[\exp\left(\frac{A_{\tilde{\nu}^*}(s,a)}{1+\alpha} - 1\right)\right]$$

Then, $\hat{\nu} := \tilde{\nu}^* + C^*$ satisfies

$$\mathbb{E}_{(s,a)\sim d^U}\left[\exp\left(\frac{A_{\hat{\nu}}(s,a)}{1+\alpha} - 1\right)\right]$$

$$= \mathbb{E}_{(s,a)\sim d^U}\left[\exp\left(\frac{A_{(\tilde{\nu}^*+C^*)}(s,a)}{1+\alpha} - 1\right)\right]$$

$$= \mathbb{E}_{(s,a)\sim d^U}\left[\exp\left(\frac{A_{\tilde{\nu}^*}(s,a) - (1-\gamma)C^*}{1+\alpha} - 1\right)\right]$$

$$= \mathbb{E}_{(s,a)\sim d^U}\left[\exp\left(\frac{A_{\tilde{\nu}^*}(s,a)}{1+\alpha} - \frac{1-\gamma}{1+\alpha}C^* - 1\right)\right]$$

$$= \mathbb{E}_{(s,a)\sim d^U}\left[\exp\left(\frac{A_{\tilde{\nu}^*}(s,a)}{1+\alpha} - 1\right)\right]\exp\left(-\frac{1-\gamma}{1+\alpha}C^*\right)$$

$$= \mathbb{E}_{(s,a)\sim d^U}\left[\exp\left(\frac{A_{\tilde{\nu}^*}(s,a)}{1+\alpha} - 1\right)\right]\left\{\mathbb{E}_{(s,a)\sim d^U}\left[\exp\left(\frac{A_{\tilde{\nu}^*}(s,a)}{1+\alpha} - 1\right)\right]\right\}^{-1}$$

$$= 1.$$

Furthermore, $\hat{\nu}$ is also an optimal solution of $\min_\nu \widetilde{L}(\nu; r)$ by Lemma 2. Then, by equality condition in Lemma 3,

$$L(w_{\hat{\nu}}^*, \hat{\nu}; r) = \widetilde{L}(\hat{\nu}; r) = \min_\nu \widetilde{L}(\nu; r) \leq \min_\nu L(w_\nu^*, \nu; r).$$

It means, $\hat{\nu}$ is an optimal solution of $\min_\nu L(w_\nu^*, \nu; r)$

□

**Lemma 5.** *An optimal solution* $\nu^* = \arg\min_\nu L(w_\nu^*, \nu; r)$ *is also an optimal solution of* $\min_\nu \widetilde{L}(\nu; r)$

*Proof.* From the Lemma 3,

$$\min_\nu L(w_\nu^*, \nu; r) = L(w_{\nu^*}^*, \nu^*; r) \geq \widetilde{L}(\nu^*; r).$$

Lemma 4 show that $\min_\nu L(w_\nu^*, \nu; r)$ and $\min_\nu \widetilde{L}(\nu; r)$ have the same minimum value,

$$\widetilde{L}(\nu^*; r) \leq L(w_{\nu^*}^*, \nu^*; r) = \min_\nu L(w_\nu^*, \nu; r) = \min_\nu \widetilde{L}(\nu; r)$$

holds. It means that $\widetilde{L}(\nu^*; r)$ is the minimum value of $\widetilde{L}$ and therefore, $\nu^*$ is an optimal solution of $\min_\nu \widetilde{L}(\nu; r)$. $\qquad\square$

Finally, we ready to prove the corollary:

**Corollary 1.** *Let* $\widetilde{V} = \arg\min_\nu \widetilde{L}(\nu; r)$ *and* $V = \arg\min_\nu L(w_\nu^*, \nu; r)$ *be the sets of* $\nu$. *Then the following equation holds: (We provide the proof in Appendix B.)*

$$\widetilde{V} = \{\nu + c | \nu \in V, c \in \mathbb{R}\} \tag{19}$$

*Proof.* Lemma 4 and Lemma 5 shows that the last equation holds. $\qquad\square$

## C   PROOF OF PROPOSITION 2

**Proposition 2.** *The objective $\widetilde{L}(\nu; r)$ is convex with respect to $\nu$. (The proof is in Appendix C.)*

*Proof.* For any functions $\nu, \nu'$ and constant $\lambda \in [0, 1]$, we can derive the following equality:

$$
\begin{aligned}
&A_{(\lambda\nu+(1-\lambda)\nu')}(s, a) \\
&= r(s, a) + \gamma(\mathcal{T}(\lambda\nu + (1-\lambda)\nu'))(s, a) - (\lambda\nu(s) + (1-\lambda)\nu'(s)) \\
&= r(s, a) + \gamma\lambda(\mathcal{T}\nu)(s, a) + \gamma(1-\lambda)(\mathcal{T}\nu')(s, a) - (\lambda\nu(s) + (1-\lambda)\nu'(s)) \\
&= \lambda(r(s, a) + \gamma(\mathcal{T}\nu)(s, a) - \lambda\nu(s)) + (1-\lambda)(r(s, a) + \gamma(\mathcal{T}\nu')(s, a) - \lambda\nu'(s)) \\
&= \lambda A_\nu(s, a) + (1-\lambda)A_{\nu'}(s, a).
\end{aligned}
$$

It means, $A_\nu$ is the linear function with respect to $\nu$. Furthermore, using the convexity of log-sum-exp,

$$
\begin{aligned}
&\log \mathbb{E}_{(s,a)\sim d^U} \left[ \exp(A_{(\lambda\nu+(1-\lambda)\nu')}(s, a)) \right] \\
&= \log \mathbb{E}_{(s,a)\sim d^U} \left[ \exp(\lambda A_\nu(s, a) + (1-\lambda)A_{\nu'}(s, a)) \right] \\
&\leq \lambda \log \mathbb{E}_{(s,a)\sim d^U} \left[ \exp(A_\nu(s, a)) \right] + (1-\lambda) \log \mathbb{E}_{(s,a)\sim d^U} \left[ A_{\nu'}(s, a)) \right]
\end{aligned}
$$

Therefore, $\widetilde{L}(\nu; r)$ is convex function with respect to $\nu$. $\qquad\square$

## D   POLICY EXTRACTION

By the definition of $A_\nu$,

$$
\begin{aligned}
&\exp\left( \frac{A_{(\tilde{\nu}^*+C)}(s, a)}{1+\alpha} - 1 \right) \\
&= \exp\left( \frac{A_{\tilde{\nu}^*}(s, a) - (1-\gamma)C}{1+\alpha} - 1 \right) \\
&= \exp\left( \frac{A_{\tilde{\nu}^*}(s, a)}{1+\alpha} \right) \exp\left( -\frac{1-\gamma}{1+\alpha}C - 1 \right).
\end{aligned}
$$

Therefore,

$$
\exp\left( \frac{A_{\tilde{\nu}^*}(s, a)}{1+\alpha} \right) \propto \exp\left( \frac{A_{(\tilde{\nu}^*+C)}(s, a)}{1+\alpha} - 1 \right) = \frac{d^*(s, a)}{d^U(s, a)}.
$$

# E   EXPERIMENTAL DETAILS

**Implementation detail**   For fair comparison, we use the same learning rate to train actors of BC and DemoDICE. In DemoDICE, we applied the same regularization coefficient $\alpha = 0.05$ in Equation 17 across all the tasks and dataset configurations. We implement our network architectures for BC and DemoDICE based on the implementation of OptiDICE[2]. For ValueDICE, we use its official implementation[3] without any modification to network architectures or hyperparameters. Based on DAC (Kostrikov et al., 2019), we treat terminal states as absorbing states in ValueDICE and DemoDICE.

For stable discriminator learning, we use gradient penalty regularization on the $r(s, a)$ function, which was proposed in Gulrajani et al. (2017) to enforce 1-Lipschitz constraint. To stabilize critic training, we add gradient L2-norm to the critic loss for the regularization. Detailed hyperparameter configurations used for our main experiments are summarized in Table 1.

| Hyperparameters | BC | DemoDICE |
|---|---|---|
| $\gamma$ (discount factor) | 0.99 | 0.99 |
| $\alpha$ (regularization coefficient) | - | 0.05 |
| learning rate (actor) | $3 \times 10^{-5}$ | $3 \times 10^{-5}$ |
| network size (actor) | [256, 256] | [256, 256] |
| learning rate (critic) | - | $3 \times 10^{-4}$ |
| network size (critic) | - | [256,256] |
| learning rate (discriminator) | - | $3 \times 10^{-4}$ |
| network size (discriminator) | - | [256,256] |
| gradient L2-norm coefficient (critic) | - | $1 \times 10^{-4}$ |
| gradient penalty coefficient (discriminator) | - | 10 |
| batch size | 256 | 256 |
| # of expert trajectories | 1 | 1 |
| # of training iterations | 1,000,000 | 1,000,000 |

Table 1: Configurations of hyperparameters used in our experimental results.

## E.1   EXPERIMENTAL DETAILS FOR BCND

We implement the BCND based on the BCND paper (Sasaki & Yamashina, 2021). We shuffle all demonstrations and divide them into $K$ disjoint sets $\{D^k\}_{k=1}^K$. Then, for each iteration, we perform stochastic gradient ascent to update each policy $\pi_{\theta^k}$ using $D^K$. Finally, we update reward $\hat{R}(s, a)$ and $\eta$ using

$$\hat{R}(s, a) \leftarrow \frac{1}{K} \sum_{k=1}^K \pi_{\theta^k}(a|s), \quad \forall (s, a) \in D,$$

$$\eta \leftarrow \frac{K}{\mathbb{E}_{s \sim D}[\sum_k \pi_{\theta^k}(\mu_{\theta^k}(s)|s)]}.$$

Here, we use following hyperparameters as reported in paper (Sasaki & Yamashina, 2021): We use $M = 5$ (number of iteration), $K = 5$ (number of policy / disjoint dataset), $N = 128$ (batch size), $L = 500 \times |D^k|/N$ for each $D^k$ ($k \in \{1, \ldots, K\}$, in our tasks, $L \in [3 \times 10^5, 1.1 \times 10^6]$), $\pi_\theta(a|s) = \frac{1}{K} \sum_{k=1}^K \pi_{\theta^k}(a|s)$, $\zeta = 10^{-4}$ (shared learning rate), Adam optimizer with learning rate $\zeta \times \eta$ for gradient ascent, and Gaussian policy $\pi_{\theta^k} = \mathcal{N}(\mu_{\theta^k}(s), \sigma_{\theta^k}^2)$. For Guassian policy, we use neural network with size $[100, 100]$ for $\mu_{\theta^k}$, and a trainable independent vector for $\sigma_{\theta^k}$, as reported in the paper.

---

[2] https://github.com/secury/optidice
[3] https://github.com/google-research/google-research/tree/master/value_dice

We measure the performance of algorithm for each internal iteration. In Figure 1 and Figure 2, we report the best performance and corresponding standard error of the algorithm among the $M$ internal iterations (we compute the average performance for each iteration and pick the maximum average performance as the best performance).

---

**Algorithm 1** Behavioral Cloning from Noisy Demonstrations

---

**Require:** Noisy expert demonstrations $D$, policy parameters $\{\theta_k\}_{k=1}^K$, learning rate $\zeta$
**Ensure:** Ensemble policy $\pi_\theta = \frac{1}{K}\sum_{k=1}^K \pi_{\theta^k}$
  Set $\hat{R}(s,a) = 1$ for $\forall(s,a) \in D$.
  Split $D$ into $K$ disjoint sets $\{D^1, D^2, \ldots, D^K\}$.
  **for** iteration$= 1, \ldots, M$ **do**
    **for** $k = 1, \ldots, K$ **do**
      Initialize parameters $\theta^k$.
      **for** $l = 1, \ldots, L$ **do**
        Sample a minibatch $B^k = \{(s_n, a_n)\}_{n=1}^N$ from $D^k$.
        Calculate a sampled gradient $\frac{1}{N}\sum_{n=1}^N \nabla_{\theta^k} \log \pi_{\theta^k}(s_n, a_n) \cdot \hat{R}(s_n, a_n)$.
        Update $\theta^k$ by gradient ascent with learning rate $\zeta \times \eta$
      **end for**
    **end for**
    Update $\hat{R}(s,a)$ and $\eta$
  **end for**

---

## F   Additional Experimental Results

In this section, we report additional experimental results for detailed empirical analysis of our approach. We focus three experiments as follows: (1) asymptotic performance of our approach on HalfCheetah environment for the mixed dataset we considered in Section 5.1, (2) behavior cloning with more diverse $\beta$ selections which are omitted in Figure 1, (3) ablation study on hyperparameter $\alpha$ for DemoDICE, which is appeared in Equation 17.

### F.1   Asymptotic Performance of DemoDICE on HalfCheetah

To inspect asymptotic behaviors of DemoDICE on our mixed dataset tasks in HalfCheetah environment, we extend the number of training iterations into 2 million with fixing the other hyperparameter settings. As Figure 3 shows, DemoDICE converges to the expert performance for all tasks we considered in Section 5.1.

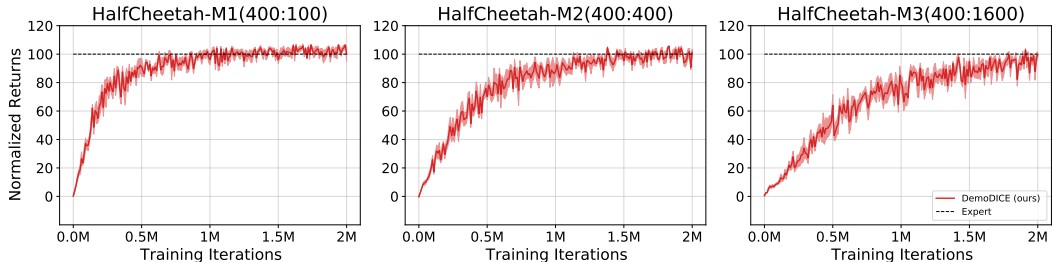

Figure 3: 1 Learning curves of DemoDICE during 2 million training iterations in HalfCheetah mixed dataset tasks (M1, M2, M3).

## F.2 BEHAVIOR CLONING PERFORMANCE

In Figure 1, we omitted performance of BC with $\beta = 0.25$ and $\beta = 0.75$ for simplicity. Figure 4 shows performance of entire BC configurations, i.e., $\beta \in \{0, 0.25, 0.5, 0.75, 1.0\}$.

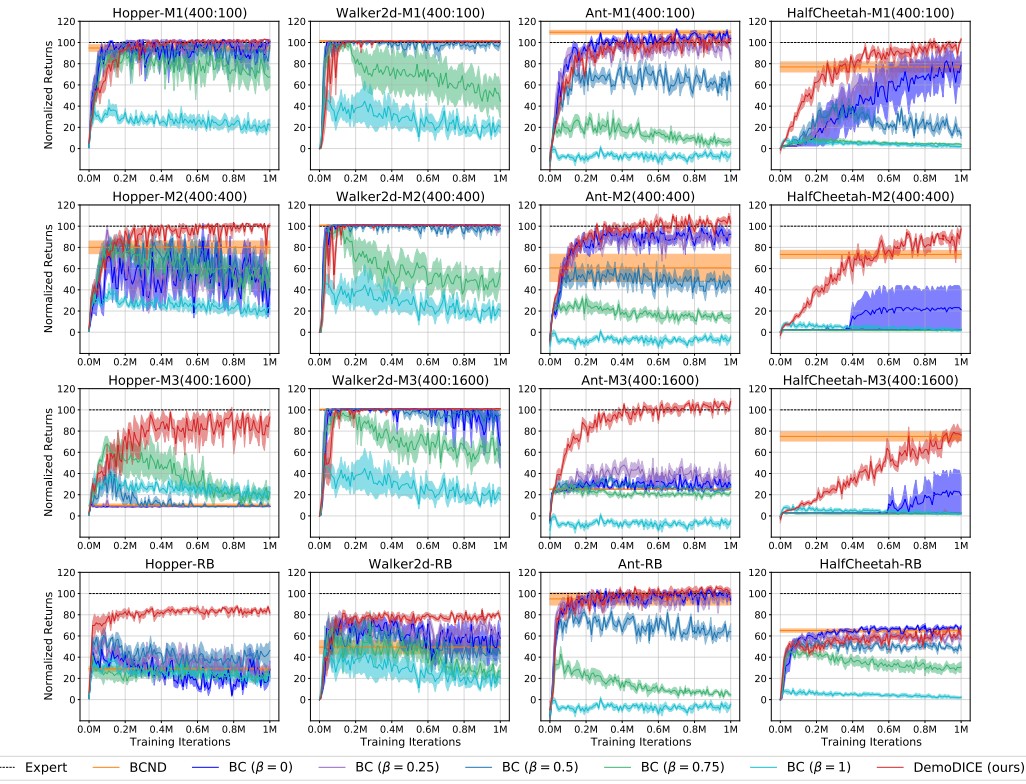

Figure 4: Performance of variants of BC with $\beta \in \{0, 0.25, 0.5, 0.75, 1.0\}$ and DemoDICE both on the mixed datasets (M1, M2, M3) and the replay buffer datasets. We plot the mean and the standard errors (shaded area) of normalized scores over 5 random seeds.

## F.3 DIFFERENT QUALITY OF IMPERFECT DEMONSTRATIONS

In this section, we construct new tasks where random trajectories in mixed dataset (see 5.1) are replaced by medium trajectories, named I1, I2, and I3. Here, all tasks share the expert demonstrations composed of the single expert trajectory. Imperfect demonstrations in each task are composed of 400 expert trajectories and medium trajectories, and the number of medium trajectories is 100 for the task I1, 400 for I2, and 1600 for I3. In Figure 5, we observed that DemoDICE performs well in these new tasks.

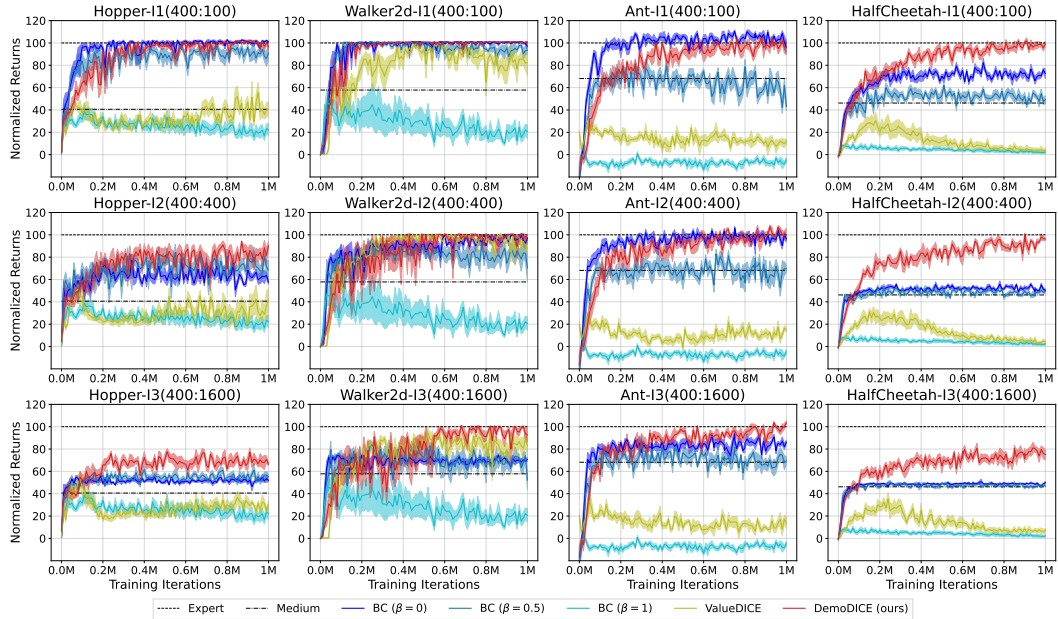

Figure 5: Performance of DemoDICE and baseline algorithms on alternative mixed-dataset tasks I1, I2, and I3. Bold lines and shaded areas indicate the means and the standard errors of normalized scores over 5 random seeds respectively.

## F.4   DIFFERENT COVERAGE OF EXPERT DEMONSTRATIONS

In this section, we introduce additional experiments with different coverages of expert demonstrations. Concretely, we constructed expert demonstrations using an increasing number of expert trajectories, namely 1, 2, 5, up to 10. Because DemoDICE already performs well in M1 and M2 tasks, we compare M3 and RB tasks for this ablation study. In this experiments, DemoDICE is not significantly better than other baselines when the expert demonstrations cover enough space. We remind the readers that when the coverage is not high, DemoDICE significantly outperforms the others as shown in Figure 1 and 2 in the main text.

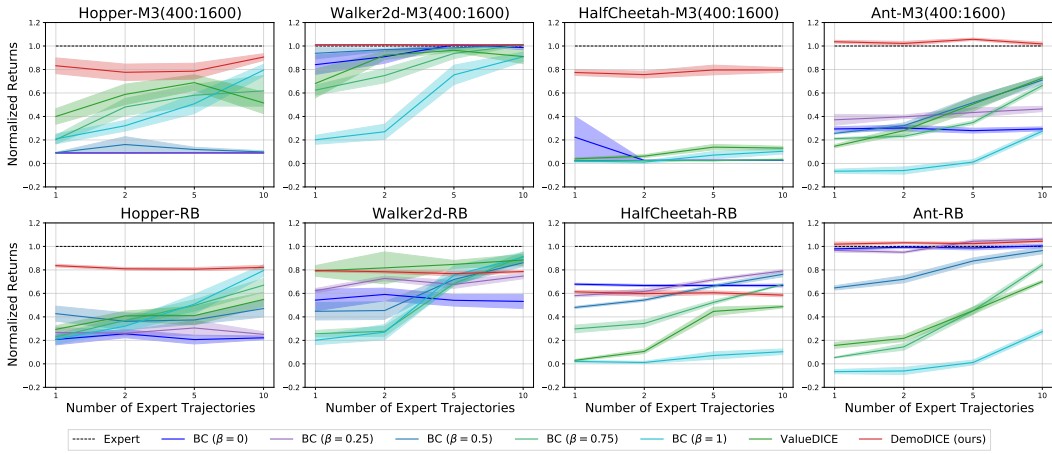

Figure 6:  We compare DemoDICE with baselines when the number of expert trajectories constituting expert demonstrations is 1, 2, 5, and 10. Each performance is measured as a normalized score averaged by the last 5 evaluations during training. Bold lines and shaded areas indicate the means and the standard errors of converged performances over 5 random seed respectively.

### F.5 Ablation Test on Hyperparameter $\alpha$

As we can notice in Equation 17, DemoDICE introduces a hyperparameter $\alpha$, which is a regularization coefficient. In this section, we aim to study how does the choice of $\alpha$ effect to training processes and performance of our algorithm. We select $\alpha \in \{10, 1, 0.5, 0.1, 0.05, 0\}$ and train on M3 datasets and replay buffer datasets by preserving the other settings same as before. Figure 7 describes the results of ablation study. We point out that DemoDICE shows almost the same performance in when $\alpha \in [0, 0.1]$.

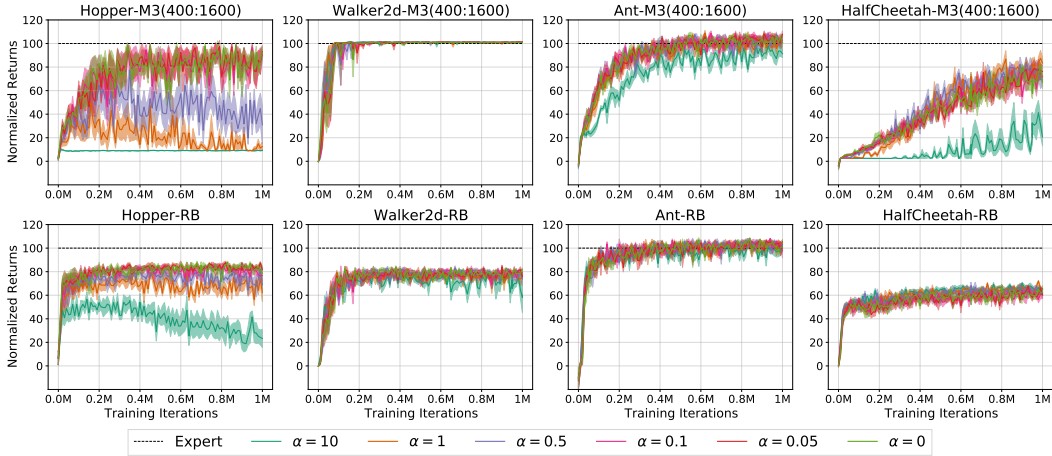

Figure 7: Performance of DemoDICE by varying $\alpha \in \{10, 1, 0.5, 0.1, 0.05, 0\}$ on the mixed datasets M3 and the replay buffer datasets. Bold lines and shaded areas indicate the means and the standard errors (shaded area) of normalized scores over 5 random seeds respectively.

### F.6 ABLATION TEST ON PURE OFFLINE IMITATION LEARNING

In this section, we compare DemoDICE and BC using only expert demonstrations. In Figure 8, we observe that DemoDICE and BC show similar performance. In pure offline IL, DemoDICE essentially tries to find a stationary distribution that minimizes $D_{KL}(d||d^E)$ on the MLE MDP constructed by $D^E$. Let $\hat{d}^E$ be the empirical stationary distribution of $d^E$. Then, the optimal solution is trivially given by $d^* = \hat{d}^E$ with $D_{KL}(d||\hat{d}^E) = 0$ since $\hat{d}^E$ itself is a valid stationary distribution on the MLE MDP. Finally, extracting a policy from $d^E$ reduces to a simple behavior cloning: $\min_\pi -E_{(s,a)\sim d^E}[\log \pi(a|s)]$, which concludes that DemoDICE would not be better than BC in the pure offline IL setting.

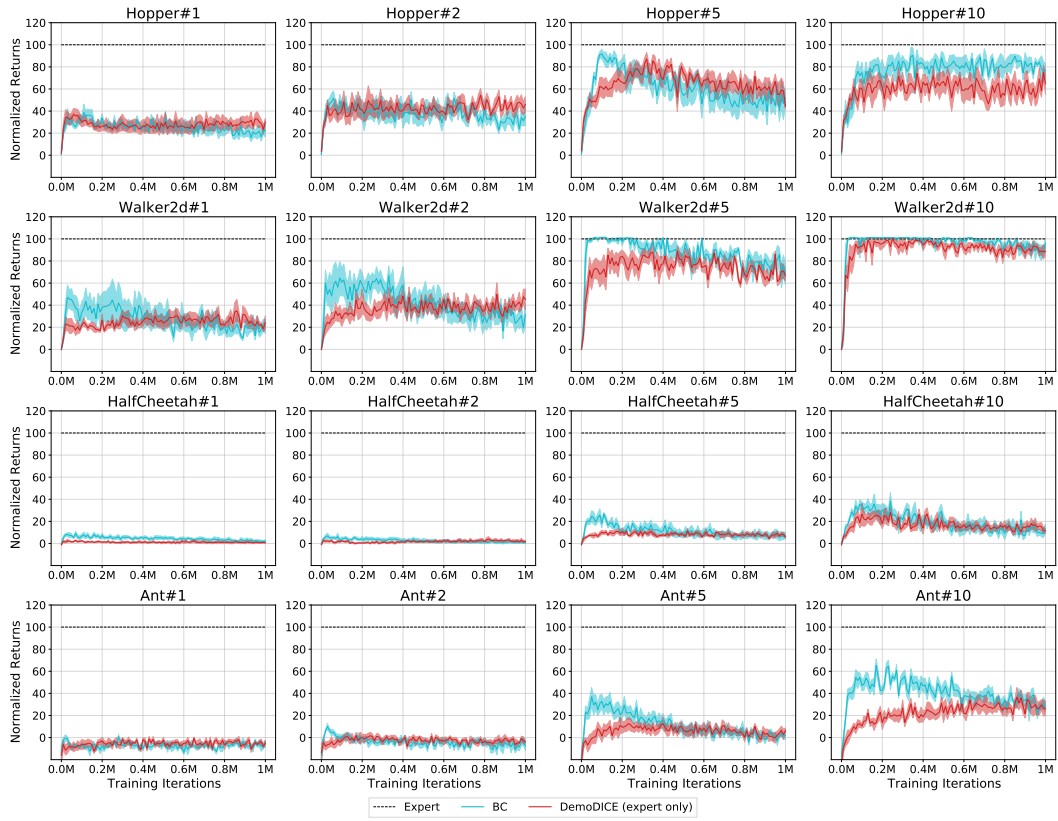

Figure 8: Performance of DemoDICE and BC with only expert demonstrations. For each environment, we construct expert demonstrations using an increasing number of expert trajectories, namely 1, 2, 5, up to 10. Bold lines and shaded areas indicate the means and the standard errors of normalized scores over 5 random seeds respectively.

# G    Detailed Analysis of Empirical Results

In this section, we provide additional discussions about empirical results.

First, M1 is an easy task where most of the trajectories are from the expert, so some baselines also succeed to achieve good performance.

| Environment | All trajectories | $> 100$ | $> 90$ | $> 80$ | $> 70$ | $> 60$ | $> 50$ |
|---|---|---|---|---|---|---|---|
| Hopper | 3514 | 0 | 117 | 176 | 217 | 281 | 385 |
| Walker2d | 1887 | 0 | 0 | 325 | 564 | 638 | 693 |
| Ant | 1318 | 333 | 479 | 576 | 629 | 672 | 727 |
| HalfCheetah | 999 | 0 | 0 | 0 | 143 | 467 | 689 |

Table 2: Statistics of trajectories in `full_replay-v2` in D4RL datasets.

To discuss about RB tasks, we show the normalized score statistics of trajectories in the replay buffer in the Table 2. For each environment, we counted the number of trajectories that have a normalized score greater than $R$ as $> R$, so the numbers above are cumulative counts.Thus, this table suggests natural upper bound of performances of imitation learning algorithms, e.g. it would be very hard to score around 100 for HalfCheetah task.

In the Ant-RB task, we have enough good trajectories, especially $> 100$, so both DemoDICE and BC($\beta = 0$) recover expert policy.

In the HalfCheetah-RB task, there is no trajectory with $> 80$, so DemoDICE eagerly distinguishes the non-expert trajectories from expert ones, throwing away non-expert demonstrations. Thus it overall behaves like BC($\beta = 0$).

In the Walker2d-RB task, both ValueDICE and DemoDICE show similar performances at around 80 at the end. Looking at the statistics above, since the best trajectory in RB has its normalized score less than 90, we believe that ValueDICE and DemoDICE both achieved reasonable performance levels.

