# OpenReview forum: "DemoDICE: Offline Imitation Learning with Supplementary Imperfect Demonstrations"
_ICLR.cc/2022/Conference — ICLR 2022 Poster_

### Official Review · Reviewer_SEq7 · 2021-10-29

**Correctness:** 4
**Technical Novelty And Significance:** 2
**Empirical Novelty And Significance:** 3
**Recommendation:** 8
**Confidence:** 4

**Main Review:**

Strengths:

-- The proposed setting is practical and important.

-- The proposed algorithm is novel (although heavily derives from existing work).

-- The empirical results & baselines are comprehensive.

-- The writing is clear and easy to follow.

Weaknesses/Questions:

-- The 3-stage process in the algorithm is arguably more complex than baselines (e.g., BC is just a single objective, ValueDICE is a single min-max objective) and can present challenges in practice w.r.t. hyperparameter tuning of each stage.

-- The derivations heavily borrow from existing work (e.g., OptiDICE, AlgaeDICE). At times, the derivations seem to be more complicated than they need to be. For example, when introducing the log-expected-exponent, it is presented as a very involved multi-step derivation. I think it could be simplified by just noting that the Fenchel dual of KL when inputs are restricted to the simplex is exactly the log-expected-exponent function; see Section 5.3.1 in https://arxiv.org/abs/2001.01866 (in fact, this existing tutorial includes a lot of derivations similar to DemoDICE).

-- I have missed this in the paper, but I'd be curious to know how DemoDICE performs in the pure imitation learning setting (no imperfect demos) compared to the baselines.

-- You may want to include a reference to https://arxiv.org/abs/2102.13185 which also proposes an IL algorithm that is composed of a 1st stage that does discriminator-based density ratio estimation and then uses that density ratio as a reward for a later DICE-like stage.

**Summary Of The Paper:**

The paper considers the problem of imitation learning, and specifically the setting where a small number of expert demonstrations is paired with some amount of suboptimal data. To tackle this setting, the paper proposes to optimize a convex constrained optimization, with linear variables denoting d^pi, linear constraints expressing the MDP transitions, and an objective composed of a linear combination of KL from the expert and KL from the suboptimal data. The paper presents derivations transforming this convex constrained optimization to a practical, unconstrained objective. The final algorithm is composed of three stages: (1) train a discriminator to get a density ratio estimator; (2) use the density ratios as a reward in a separate log-expected-exp objective (similar to REPS or ValueDICE objectives) to learn a different density ratio w; (3) extract a policy from w using weighted max likelihood training on the suboptimal data. The algorithm is evaluated on a variety of simulated robotics (mujoco) environments, and shows favorable results compared to baselines.

**Summary Of The Review:**

Overall the paper is a good submission, and I would recommend accepting it. I hope the authors can incorporate my feedback into the final version.

---

> ### Author Response · Authors · 2021-11-21
> **Response to Reviewer SEq7**
>
> Thank you for your constructive and thoughtful feedback.
>
> 1. [Hyperparameter tuning] DemoDICE is seemingly more complex than other baseline methods, but this is to leverage supplementary imperfect demonstrations. Regarding implementation details, we have a comparable number of hyperparameters with ValueDICE.
>
>     First, both DemoDICE and ValueDICE have a hyperparameter $\alpha$ for KL-regularization and mixture of samples, respectively. We also point out that the performance of DemoDICE is not sensitive to the choice of $\alpha$ in our experiment reported in Appendix F.5.
>
>     Also, DemoDICE uses two regularizers for the discriminator and $\nu$ while ValueDICE uses two regularizers for $\nu$ and $\pi$.
> DemoDICE has only one additional learning rate compared with ValueDICE, but we set the same learning rate for the discriminator and $\nu$.
>
>     Finally, ValueDICE and DemoDICE use neural networks with the same hidden layers (two hidden layers with 256 hidden units for each). This architecture has been widely used in many RL and IL algorithms.
>
> 2. [Derivation] We appreciate your suggestion. We revised the proof in Appendix A based on the Fenchel duality. In summary, based on the primal and dual problem in Section 2.2 in [3.1], we derive $\tilde L(\nu;r)$ similarly to the derivation in Section 5.3.1 in [3.1].
>
> 3. [Pure offline IL] We added new experimental results for the pure offline IL setting in Appendix F.6. In Figure 8, we observe that DemoDICE and BC show similar performance.
> Please note that DemoDICE does not necessarily have advantage over BC in the pure offline IL setting since they are equivalent.
> In the pure offline IL, DemoDICE essentially tries to find a stationary distribution that minimizes $D_\text{KL}(d || d^E)$ on the MLE MDP constructed by $D^E$.
> Bearing this in mind, the optimal solution is trivially given by $d^* = d^E$ with $D_\text{KL}(d || d^E)=0$ since $d^E$ itself is a valid stationary distribution on the MLE MDP. Thus, extracting a policy from $d^E$ reduces to a simple behavior cloning: $\min_{\pi} - E_{(s,a) \sim d^E}[ \log \pi(a|s) ]$
>
> 4. [Related work] Thank you for your recommendation. We revised the related work section.
>
>     Off policy learning from observations (OPOLO) [3.2] addresses off-policy IL using expert demonstrations composed only of states. It has similarities to DemoDICE in two aspects. Firstly, OPOLO adopts a discriminator to estimate a state-transition distribution ratio. Furthermore, DICE-like technique is applied to the derivation of the objective of OPOLO, which is similar to ours. Thus, based on the relation between DemoDICE and OPOLO, offline IL using expert demonstrations containing only states can be considered as a promising direction for future work.
>
> [3.1] Nachum, Ofir, and Bo Dai. "Reinforcement Learning via Fenchel-Rockafellar Duality." arXiv preprint arXiv:2001.01866, 2020.
>
> [3.2] Zhu, Zhuangdi, et al. "Off-policy Imitation Learning from Observations." NeurIPS, 2020.

---

> > ### Public Comment · ~Roy_Fine1 · 2022-11-16
> > **Thanks**
> >
> > Thanks for the feedback. Do you require the greatest essay writing service since you are unable to accomplish your college assignments? The finest post for you is at https://writinguniverse.com/free-essay-examples/north-korea/. The essay writing samples on this website are free to use. With this, you may compose an essay on any subject. Additionally, you can let your friends know about this page.

---

> > > ### Public Comment · ~Thomas_Moran1 · 2023-08-21
> > > **The three-country**
> > >
> > > The three-country framework allows specific attention to be paid to the international transmission of fiscal policy between trading partners that have different production and indebtedness profiles. https://dinosaur-game.io

---

### Official Review · Reviewer_ERsJ · 2021-11-01

**Correctness:** 3
**Technical Novelty And Significance:** 2
**Empirical Novelty And Significance:** 3
**Recommendation:** 8
**Confidence:** 3

**Main Review:**

### Strengths
The proposed problem is a natural and practical extension of the offline IL problem. This is a nice step towards practical applications of IL. The derivation of the method is also grounded. While the derivation steps are based on existing work, the stability improvement by using a soft-max surrogate and its accompanying proofs are good additions.
### Weaknesses
1) Related work

It is interesting to see that the final policy learning objective in equation 21 essentially reweighs behavioral cloning (BC) objective based on the advantage function. This result makes DemoDICE closely related to [1] which reweighs GAIL objective based on the advantage function and [2] which learns from imperfect demonstrations by reweighting a BC objective. These recent works should be discussed in the paper.

2) Experiments

Experiments are conducted sufficiently to demonstrate the effectiveness of DemoDICE on continuous-control benchmarks. Though, there are only two baseline methods (BC and ValueDICE) and more baselines should be evaluated (e.g., the method in [2]).

3) Clarity

The paper is overall well written and well organized. However, there are unclear statements that need further clarification:

  - In Section 3.3, why cannot $w^\star$ be computed from $\tilde{\nu}^\star$? In my understanding, $\tilde{\nu}$ is a neural network that minimizes equation (18) and the advantage function can be computed directly from it. With these, it should be straightforward to compute $w^\star(s,a) = \exp(\frac{A_{\tilde{\nu}^\star}(s,a)}{1+\alpha}-1)$ on state-action samples drawn from the dataset $D^U$ to reweight the objective.

  - I do not understand the sentence "Based on the weighted BC, we simply extract the policy without training any additional network.". It is indeed  possible to find a non-parametric $\pi$ from equation (21). However, such $\pi$ only defines action probabilities on states observed in the datasets and cannot be used in the environment. At the same time, the appendix does mention about a neural network for the actor. So I do not understand which  "additional network" the above sentence refers to.

  - Proposition 1 requires $\nu$ to be in a large family of functions. Are there restrictions of $\nu$? Lemma 1 and 2 seem to hold for any function $\nu$, so I think Proposition 1 should hold for any $\nu$.

### Questions

1) It is intriguing to see that regularized BC does not work in the experiments but DemoDICE does given that BC is also a distribution-matching approach (without matching the state-marginals). Does the superior performance of DemoDICE come from matching the state-marginals or from the dual-program optimization?

2) In theory, the policy function can be extracted from state-action distribution exactly with $\pi(a|s) = \frac{d^\pi(s,a)}{\int_A d^\pi(s,a) \text{d}a}$. The integration is intractable in general for continuous control tasks. However, for Mujoco tasks the action-space is bounded in $[-1,1]$. Is computing the exact policy function possible in DemoDICE with the knowledge of action-space's bounds?

[1] Yunke Wang, Chang Xu, Bo Du, and Honglak Lee. Learning to Weight Imperfect Demonstrations. ICML, 2021.

[2] Fumihiro Sasaki and Ryota Yamashina. Behavioral Cloning from Noisy Demonstrations. ICLR, 2021.



**Summary Of The Paper:**

The paper considers an offline imitation learning (IL) problem with an addition of supplementary imperfect demonstrations. To solve this problem, the paper proposes DemoDICE which regularizes a distribution-matching objective of IL by a KL divergence between the agent distribution and a mixed of expert and imperfect distributions. DemoDICE finds an optimal state-action distribution of this regularized objective by using a dual-program technique similar to that of OptiDICE (Lee et al., 2021) for offline RL with an improvement in terms of stability. Given the optimal state-action distribution, DemoDICE extracts the expert policy by performing weighted behavioral cloning.
Empirical evaluation on Mujoco tasks with D4RL datasets show that DemoDICE can efficiently and effectively solve the offline IL problem.

### Contributions
- A new learning problem combining offline IL and IL with imperfect demonstrations.
- A new model-free offline IL method based on dual-program optimization.


**Summary Of The Review:**

I overall like the idea and execution in the paper. I rate the paper as acceptance. Still, there are issues that need to be addressed (related work, baseline methods, and clarity). Nonetheless, these issues are minor and should be straightforward to address.

** Update after rebuttal **
I read the rebuttal and the other reviews. The rebuttal and the revision address my questions and concerns. I think the paper should be accepted.

---

> ### Author Response · Authors · 2021-11-21
> **Response to Reviewer ERsJ**
>
> We thank the reviewer for the thoughtful and detailed feedback.
>
> 1. [Related work] Thank you very much for the suggested references. We revised the related work section to incorporate the comments.
>
>     Weighted GAIL (WGAIL) [2.1] uses imperfect demonstrations for imitation learning. Inspired from the previous work [2.3], the authors suggest a general weighted GAIL objective function. Unlike previous work, WGAIL predicts the weight for imperfect demonstrations without information about their qualities. However, WGAIL is an online on-policy imitation learning method, whereas our algorithm, DemoDICE, is an offline imitation learning method.
>
>     Behavioral Cloning from Noisy Demonstrations (BCND) [2.2] assumes that the expert demonstrations are sampled from a noisy expert policy. BCND learns ensemble policies with a weighted BC objective, where the weight is policy learned by weighted BC in the previous internal iteration. As the learning progresses, the mode of the learned policy moves towards that of the noisy expert policy. Thus, the learned policy tends to converge to a non-expert policy when the non-expert trajectories comprise the majority of demonstrations.
>
> 2. [Experiments] We added the BCND results in Figures 1 and 2. We use all demonstrations as noisy expert demonstrations and report the best performance among the 5 internal iterations of BCND. In Figure 1, we can observe that the normalized score results tend to decrease as we decrease the overall qualities of imperfect demonstrations (from M1 to M3), since the BCND algorithm is prone to converge to a non-expert policy when the non-expert trajectories form the most of noisy expert demonstrations, due to the nature of the aforementioned algorithm.
>
> 3. [Clarity]
>
>     (1) With $\nu^* \in \arg\min_\nu L(w_\nu^*,\nu;r)$, it is possible to compute $w^*$ directly. However, with the surrogate objective, $\tilde\nu^* \in \arg\min_\nu \widetilde{L}(\nu;r)$, we can obtain an unnormalized ratio (i.e., $c\cdot w^*(s,a)$ for a constant $c$), which is enough to perform importance sampling with self-normalization. We revised the explanation of policy extraction
>
>     (2) Yes, we train the policy network. However, we want to emphasize that we do not use other additional networks such as networks for $w$ as in [2.4] to learn the policy network. Thank you for pointing out the ambiguity. We revised that sentence.
>
>     (3) You are right. We revised the proposition.
>
> Questions
>
> 1. We think that the state-marginal matching was the most crucial for the performance. The dual-program optimization also helped to obtain a stable and tractable optimization for the offline setting.
>
> 2. We could extract the policy as suggested if we had access to $d^\pi$, but it is, unfortunately, unavailable in the offline setting. The proposed integration is intractable in general. However, we believe that the integration is amenable to importance sampling if we know the bounds of the action space. For example, we could introduce a network $w_\phi$ and train to match $\tilde w(s,a)$. Then, we can sample actions from $w_\phi$ using importance sampling, without resorting to the formula that requires $d^\pi$.
>
>
> [2.1] Yunke Wang, Chang Xu, Bo Du, and Honglak Lee. "Learning to Weight Imperfect Demonstrations." ICML, 2021.
>
> [2.2] Fumihiro Sasaki and Ryota Yamashina. "Behavioral Cloning from Noisy Demonstrations." ICLR, 2021.
>
> [2.3] Wu, Yueh-Hua, et al. "Imitation Learning from Imperfect Demonstration." ICML, 2019.
>
> [2.4] Lee, Jongmin, et al. "OptiDICE: Offline Policy Optimization via Stationary Distribution Correction Estimation." ICML, 2021.

---

> ### Public Comment · ~heritage_ericsson1 · 2023-10-30
> **The log anticipated exponent**
>
> The log anticipated exponent is introduced after a lengthy and convoluted multi-step derivation. When inputs are constrained to the simplex, the Fenchel dual of KL is identical to the log-expected-exponent function (see Section 5.3.1 i for details).
> https://phrazle.co

---

### Official Review · Reviewer_5YKV · 2021-11-02

**Correctness:** 4
**Technical Novelty And Significance:** 3
**Empirical Novelty And Significance:** 2
**Recommendation:** 8
**Confidence:** 3

**Main Review:**

Strength: the paper studies an important problem for imitation learning with only offline data. Several tricks in equation 11, 21 helps to improve the method over baselines. Converting the original minimax problem into a minimization problem helps to stabilize the training. Comparing with BC and ValueDICE demonstrates that the proposed DemoDICE can achieve a better performance in several MuJoCo environments. Especially, it is important to have the results of different number of combinations of expert/imperfect demonstrations.

Weakness:
The experiments part may still need further work.
1. In figure 1, the half cheetah environment results seem to be unfinished. The curves are not converging yet especially for M3. It would be great to see the results when the curves converge.
2. The authors mentioned that the supplementary demonstrations are imperfect, containing expert or near-expert demonstrations. However, the experiments section used either expert or random demonstrations. It would be great to see the performance of the algorithm when sub-optimal demonstrations are provided instead of totally random trajectories. According to proposal of the paper, the work is to tackle problems where expert demonstrations are not diverse enough. Is it possible to demonstrate how the performance of the proposed method will change when the expert demonstrations are of different coverage of the state/action space? For example, intuitively, when the expert demonstrations cover enough space, the difference between BC, ValueDICE and DemoDICE may not be that significant, while when the expert demonstrations is of very limited coverage,DemoDICE will be much better than BC and ValueDICE.
3. In the results, it seems that the more imperfect demonstrations given the same perfect demonstrations, all methods tend to have worse performance (it appears that M1,M2,M3 are using the same number of expert demos but M3 are given much more bad demos). If that is the case, why include bad demonstrations in the data given that only using good demos can achieve a better performance?
4. In M1 task in Hopper, Walker2d, Ant, the proposed DemoDICE isn't better than the baseline BC with beta=0. Also in the case, where the RB dataset is used, DemoDICE is not better than BC for beta=0 in ant and half cheetah. It is also worse than ValueDICE in Walker2d. Is there further explanation why this happens?

**Summary Of The Paper:**

This work presents an approach that learns a policy from expert demonstrations data in an offline setting. It is trying to solve an important problem in imitation learning: the expert demonstration may not cover the entire state/action space pretty well, and many existing algorithms require further interaction with the environment. To solve the problem of distribution shift when expert data is not diverse enough, the method proposes to introduce a large number of supplementary demonstrations of various qualities. In this way, the proposed algorithm DemoDICE does not require any on-policy samples. The method adapts the OptiDICE method for its own purpose by using a transformed objective function. The theoretical analysis and empirical experiments demonstrate the improvement of the method over BC and ValueDICE.

**Summary Of The Review:**

Overall the paper is of good quality and it studies an important problem. The proposed method formulation makes sense. However, the experiments part need improvement and clarification.

Post rebuttal: I read the authors' rebuttal and appreciate all the feedbacks. I hope the author could incorporate the suggestions by the reviewers to improve the overall quality of the paper. I raised my rating for the rebuttal and the paper has a nontrivial contribution to the field.

---

> ### Author Response · Authors · 2021-11-21
> **Response to Reviewer 5YKV**
>
> We greatly appreciate your thoughtful feedback.
>
> 1. [Convergence in HalfCheetah] We presented the converged performance of DemoDICE for HalfCheetah in Appendix F.1. DemoDICE successfully recovers the expert policy's performance at its convergence in all of HalfCheetah-M1, M2, and M3.
>
> 2. [Ablation study] We added new tasks in Appendix F.3 where random trajectories are replaced by medium trajectories, named I1, I2, and I3. Here, all tasks share the expert demonstrations composed of the single expert trajectory. Imperfect demonstrations in each task are composed of 400 expert trajectories and medium trajectories, and the number of medium trajectories is 100 for the task I1, 400 for I2, and 1600 for I3. In Figure 5, we observed that DemoDICE performs well in these new tasks.
>
>     In Appendix F.4, we added experiments with different coverages of expert demonstrations. Concretely, we constructed expert demonstrations using an increasing number of expert trajectories, namely 1, 2, 5, up to 10.  As the reviewer expected, DemoDICE is not significantly better than other baselines when the expert demonstrations cover enough space. We remind the reviewer that when the coverage is not high, DemoDICE significantly outperforms the others as shown in Figure 1 in the main text.
>
> 3. [Imperfect Demonstrations] In this paper, we aim to solve an offline IL problem with only a few expert but many non-expert or imperfect demonstrations, especially when the qualities of those imperfect demonstrations are **unknown**. Our hypothesis is that DemoDICE would perform well in such a scenario, being able to identify high-quality imperfect demonstrations automatically and exploiting them to learn a good policy. M1, M2, and M3 tasks are designed to test this hypothesis. We assume that the imitation learning agent does not know a priori the quality of each imperfect demonstration as well as the variation of the qualities of those demonstrations.
>
>     Thus, to get benefitted from these imperfect demonstrations, the agent should be able to learn how to estimate the importance of the demonstrations. This capability of the agent is tested by our tasks M1, M2, M3, which imperfect demonstrations have different variations in terms of quality.
>
> 4. [Analysis of Empirical Results]
>
>     >“In M1 task in Hopper, Walker2d, Ant, the proposed DemoDICE isn't better than the baseline BC with beta=0.”
>
>     M1 is a relatively easy task where most of the trajectories are from the expert. This explains why some baselines also succeed to achieve good performance.
>
>     >“Also in the case, where the RB dataset is used, DemoDICE is not better than BC for beta=0 in ant and half cheetah.”
>
>     | Env | All | >100 | >90 | >80 | >70 | >60 | >50 |
> |-------------|------|------|-----|-----|-----|-----|-----|
> | Hopper | 3514 | 0 | 117 | 176 | 217 | 281 | 385 |
> | Walker2d | 1887 | 0 | 0 | 325 | 564 | 638 | 693 |
> | Ant | 1318 | 333 | 479 | 576 | 629 | 672 | 727 |
> | HalfCheetah | 999 | 0 | 0 | 0 | 143 | 467 | 689 |
>
>    To discuss RB tasks, we show the normalized score statistics of trajectories in the replay buffer in the table above. For each environment, we count the number of trajectories that have normalized scores greater than R as >R, so the numbers above are cumulative counts. Thus, this table suggests a natural upper bound of the performances of imitation learning algorithms, e.g. it would be very hard to score around 100 for the HalfCheetah task.
>
>    In the Ant-RB task, we have a sufficient number of good trajectories, especially >100, so both DemoDICE and BC($\beta=0$) recover the expert policy.
>
>    In the HalfCheetah-RB task, there is no trajectory with >80, so DemoDICE eagerly distinguishes the non-expert trajectories from expert ones, neglecting non-expert demonstrations. Overall, DemoDICE performs on par with BC($\beta=0$).
>
>     >“It is also worse than ValueDICE in Walker2d.”
>
>     When the training step equals 1M, both ValueDICE and DemoDICE show similar performances at around 80. From the statistics in the above table, since the best trajectory in RB has its normalized score less than 90, we would like to argue that both ValueDICE and DemoDICE achieve reasonable performance levels.

---

### Author Response · Authors · 2021-11-21
**General Response**

We thank all the reviewers for their constructive feedback and comments. The improvements and modifications in the revision are summarized below:

[Experiments]

- We added a new baseline, Behavioral Cloning from Noisy Demonstrations (BCND) [1] in Figure 1 and Figure 2.
- We constructed new tasks I1, I2, and I3 using D4RL medium-v2 instead of random trajectories similar to M1, M2, and M3, respectively. The results are in Appendix F.3.
- We added an ablation study for the coverage of expert demonstrations. The results are in Appendix F.4.
- We added pure offline IL (without imperfect demonstrations) experiments in Appendix F.6.

[Writing]

- We revised the related work section.
- We revised the proof of Proposition 1 in Appendix A based on the Fenchel duality.
- We provided a more detailed analysis of empirical results in Appendix G.

If you have any additional questions or concerns about our response, we are more than happy to answer them.

[1] Fumihiro Sasaki and Ryota Yamashina. "Behavioral Cloning from Noisy Demonstrations." ICLR, 2021.

---

### Public Comment · ~Ziniu_Li1 · 2022-02-09
**Questions about DemoDICE**

Dear authors,

Thanks for your interesting paper. After reading, I have several questions about your paper and hope you can help me.

[Objective] From the literature, [1] [2] showed that BC is minimax optimal in the offline setting, which implies no method is better than BC in the worst case. Furthermore, [3] pointed out that compared with BC, state-action distribution matching methods does not enjoy benefits in the offline setting because they cannot leverage the transition function. In particular, [3] provided evidence that ValueDice reduces to BC in the offline setting and two methods indeed have similar empirical performances. These works are closely related to DemoDICE.

In my understanding, the objective of DemoDice in (4) is the same as ValueDice except that (4) has an additional term of KL for the non-expert dataset. Thus, I have a similar question with Reviewer ERsJ: what matters for the superior performance of DemoDice?  I have seen your answer that ''the state-marginal matching was the most crucial''. However, your claim seems not supported by the experiment results in Appendix F.6 (i.e., the pure offline imitation learning setting). In particular, results in Appendix F.6 would suggest that BC and DemoDice have similar performances. According to the mentioned works, I tend to believe that there are no fundamental differences between DemoDice and the weighted BC in the offline setting. Could you share some ideas?

[Policy Extraction 1] In DemoDice, policy extraction is based on (18) rather than the max-return RL. I cannot see that the optimal solution to (18) is identical to the one in (4). Could you clarify this point? In particular, why can (self)-weighted BC perform state-action distribution matching? If my understanding is correct, Corollary 1 does not show this relation.

[Policy Extraction 2] In Appendix F.6, you argue that "extracting a policy from $d^E$ reduces to a simple behavior cloning". I believe this claim is wrong. Otherwise, there is no difference between state-action distribution matching methods and behavior cloning. Indeed, many papers validate two objectives are quite different; see e.g., [1] [2] [3] [4].

[BC performance] We should expect BC ($\beta=1$) to perform well because it performs imitation only on the expert dataset. However, this conjecture is not supported by the results in Figure 1. Could you explain more about this point?

I also notice that M1, M2, and M3 should have the same expert dataset. However, the performances of BC ($\beta=1$) on M1, M2, and M3 are different. Could you explain this point?

[Dataset] I am curious about which type of dataset you have used in the experiments: complete trajectories or subsampled trajectories? In particular, [3] showed that ValueDice does not work in the subsampled trajectories case, which suggests that the Dice-based objective is hard to optimize. Thus, I wonder about the performance of DemoDice in the subsampled trajectories case if applicable.

Thanks for your consideration and looking forward to your reply.

[1] Rajaraman, Nived, et al. "Toward the fundamental limits of imitation learning." Advances in Neural Information Processing Systems 33 (2020): 2914-2924.

[2] Tian Xu, Ziniu Li, and Yang Yu. "Error Bounds of Imitating Policies and Environments for Reinforcement Learning." IEEE Transactions on Pattern Analysis and Machine Intelligence (2021).

[3] Ziniu Li, Tian Xu, Yang Yu and Zhi-Quan Luo. "Rethinking valuedice: Does it really improve performance?" arXiv, 2202.02468, 2022.

[4] Ghasemipour, Seyed Kamyar Seyed, Richard Zemel, and Shixiang Gu. "A divergence minimization perspective on imitation learning methods." Conference on Robot Learning. PMLR, 2020.

---

> ### Public Comment · ~Geon-Hyeong_Kim2 · 2022-02-28
> **Response to the questions**
>
> Thank you for the meaningful discussion.
>
> [objective]
>
> We agree that when $\alpha=0$ DemoDICE reduces to BC in the pure offline IL setting where we have only few expert demonstrations, without imperfect demonstrations.
> In contrast, distribution matching allows us to use imperfect demonstrations in a principled manner.
> Therefore, DemoDICE can achieve better performance than BC in offline IL when supplementary imperfect demonstrations are given.
>
> In addition, although both ValueDICE and DemoDICE are derived from the KL-divergence matching, there are two main differences.
> First, ValueDICE solves a nested max-min problem, which is susceptible to numerical instability.
> In contrast, DemoDICE solves a single convex minimization problem, which secures the stability.
> Second, ValueDICE uses $\nu(s,a)$, which can suffer from overestimation induced by using out-of-distribution action values.
> Indeed, many offline RL algorithms aim to avoid such overestimation.
> On the other hand, DemoDICE uses $\nu(s)$ without requiring any out-of-distribution action values.
> We strongly believe that these differences led to better performance in offline settings.
>
> In summary, DemoDICE can effectively use imperfect demonstrations as opposed to BC.
> Compared to ValueDICE, DemoDICE solves the single minimization problem and uses $\nu(s)$ instead of $\nu(s,a)$.
>
> [Policy Extraction 1]
>
> First, we aim to solve constrained optimization (5-7) that directly optimizes stationary distribution $d$ instead of policy $\pi$.
> Then, we derive the dual form of the optimization with dual variable $\nu$.
> Thus, we need to extract a policy from $\nu^*$ based on the relationship between $\pi^*$ and $\nu^*$, which is described in Corollary 1.
> This connection allows us to extract policy $\pi^*$ from $\nu^*$ using self-normalized importance sampling.
> Although this estimate is biased, increasing the number of samples to infinite enforces the bias to converge to zero.
>
> [Policy Extraction 2]
>
> As you mentioned in [objective], we believe that distribution matching is similar to BC in the pure offline IL setting.
> However, in the online setting, distribution matching can leverage environment interaction whereas BC cannot.
> Compared to BC, the main advantage of distribution matching is the effective usage of the information including the additional interaction.
> We would like to emphasize that DemoDICE effectively uses supplementary imperfect demonstrations in a principled manner.
>
> [BC performance]
>
> First of all, please note that BC ($\beta=1$) shows the exactly same performance among the tasks M1, M2, and M3, which clearly shows that BC poorly performs across most of the domains given only one expert trajectory.
> On the other hand, thanks to effective utilization of the supplementary imperfect demonstrations, DemoDICE successfully learns optimal policy based on the single expert trajectory.
>
> [Dataset]
>
> We use complete trajectories and do not consider subsampling.
> However, we agree that designing an algorithm that robustly performs in the case of subsampling trajectories is also an interesting research direction.
> We plan to release our implementation so that one can evaluate our algorithm in such case.

---

> > ### Public Comment · ~Ziniu_Li1 · 2022-03-01
> > **Thanks for Reply**
> >
> > Dear Geon-Hyeong,
> >
> > Thanks for your detailed answers, which improves my understanding of DemoDICE. I have some additional (easy) questions and hope you can help me again.
> >
> > [objective] I agree with your claim about the (implicit) information introduced by the supplementary dataset is important. This additional dataset can be used to recover the transition dynamics so that the state-action matching in DemoDICE can work even in the offline setting. In contrast, simple BC and its variants cannot leverage this information. This setup is different from the pure offline imitation learning in [3].
> >
> > [Policy Extraction 1] I understand that there exists bias with an empirical self-normalized importance sampling. However, I am not sure whether this kind of approach is better or worse than the typical max-return RL method. Do you have plans to investigate this direction in the future?
> >
> > [Policy Extraction 2] Thanks for the clarification. I agree with your explanations.
> >
> > [BC performance] I am sorry that I made a mis-observation about BC($\beta=1$). I would like to further comment that with small weight decay, BC($\beta=1$) performs well (especially for Hopper and HalfCheetah) in the single expert trajectory case; see [3]. Without any regularization, BC (and maybe other methods) would suffer the overfitting in this low data regime. It would be great to consider this variant in the offline setting.
> >
> > [Dataset] Thanks for your clarification and willingness to open-source.
> >
> > Thanks for your time and consideration.

---

> > > ### Public Comment · ~Geon-Hyeong_Kim2 · 2022-04-15
> > > **Thank you for your comment**
> > >
> > > Dear Ziniu,
> > >
> > > [Policy extraction 1] Please note that DemoDICE solves the distribution matching problem by optmizing $\nu$ in (17), which is equivalent to solving (5-7).
> > > Considering that policy is not involved in optimizing (17) but extracted by (20) from the obtained $\tilde{w}_{\tilde{\nu}^*}$, we are not confident on if direct comparison between the policy extraction and max-return RL is proper.
> > > Nevertheless, we also found that comparing the pros and cons of solving offline IL via max-return and DICE can be an interesting research direction.
> > >
> > > [BC performance] We appreciate your suggestion.
> > > We still observe that BC fails to show performance improvement even when weight decay is applied to BC as described in [3] in our additional experiment

---

### Decision · Program_Chairs · 2022-01-20

**Decision:**

Accept (Poster)

**Comment:**

The paper presents a method for learning sequential decision making policies from a mix of demonstrations of varying quality. The reviewers agree, and I concur, that the method is relevant to the ICLR community. It is non-trivial, the empirical evaluations and theoretical analysis are rigorous, resulting in a novel method that produces near optimal policies from more readily available demonstrations. The authors revised the manuscript to reflect the reviewers' comments.